# Mechanical control of nuclear import by Importin-7 is regulated by its dominant cargo YAP

María García-García [1], Sara Sánchez-Perales[1], Patricia Jarabo[2], Enrique Calvo[3,4], Trevor Huyton[5], Liran Fu[5], Sheung Chun Ng[5], Laura Sotodosos-Alonso[1], Jesús Vázquez[3,4], Sergio Casas-Tintó [2], Dirk Görlich [5], Asier Echarri [1✉] & Miguel A. Del Pozo [1✉]

Mechanical forces regulate multiple essential pathways in the cell. The nuclear translocation of mechanoresponsive transcriptional regulators is an essential step for mechanotransduction. However, how mechanical forces regulate the nuclear import process is not understood. Here, we identify a highly mechanoresponsive nuclear transport receptor (NTR), Importin-7 (Imp7), that drives the nuclear import of YAP, a key regulator of mechanotransduction pathways. Unexpectedly, YAP governs the mechanoresponse of Imp7 by forming a YAP/Imp7 complex that responds to mechanical cues through the Hippo kinases MST1/2. Furthermore, YAP behaves as a dominant cargo of Imp7, restricting the Imp7 binding and the nuclear translocation of other Imp7 cargoes such as Smad3 and Erk2. Thus, the nuclear import process is an additional regulatory layer indirectly regulated by mechanical cues, which activate a preferential Imp7 cargo, YAP, which competes out other cargoes, resulting in signaling crosstalk.

[1] Mechanoadaptation and Caveolae Biology Laboratory. Area of Cell & Developmental Biology, Centro Nacional de Investigaciones Cardiovasculares (CNIC), Calle Melchor Fernández Almagro, 3, 28029 Madrid, Spain. [2] Instituto Cajal-CSIC, Avda. Doctor Arce, 37, 28002 Madrid, Spain. [3] Proteomics Unit. Area of Vascular Physiopathology, Centro Nacional de Investigaciones Cardiovasculares (CNIC), Calle Melchor Fernández Almagro, 3, 28029 Madrid, Spain. [4] CIBER de Enfermedades Cardiovasculares (CIBERCV), Madrid, Spain. [5] Department of Cellular Logistics, Max Planck Institute for Biophysical Chemistry, Am Fassberg 11, 37077 Göttingen, Germany. ✉email: aecharri@cnic.es; madelpozo@cnic.es

Mechanical cues play an important role during embryogenesis, tissue homeostasis, cell proliferation and differentiation, and gene expression[1–6]. These cues are initially sensed at the plasma membrane itself and its mechanosensitive proteins[7–10] which leads to changes in the structural components of the cell, including cytoskeleton, nucleus, and chromatin[11–15]. A key step in this response to mechanical cues (also known as mechanoresponse) is the translation of these changes into gene reprogramming, a process that depends on the nuclear import of mechanoresponsive transcriptional regulators[16–19]. Despite the importance of the nuclear translocation process in conditioning gene expression in response to mechanical cues, whether the molecular determinants that control the nuclear import process are responsive to mechanical cues is not known.

Nuclear import proceeds through nuclear pore complexes that are guarded by a permeability barrier of disordered, cohesively interacting phenylalanine-glycine (FG) repeat domains (reviewed by[20]). This FG phase has sieve-like properties and restricts the passage of inert macromolecules larger than ~30 kDa. At the same time, it allows nuclear transport receptors (NTRs) to cross rapidly in an energy-dependent manner. Human cells employ ~27 different NTRs, including nuclear exporters (exportins) as well as importins[21]. Every NTR contains a RanGTP-binding motif[22] and uses Ran to regulate the interaction with its cargoes and the passage through the nuclear pore. The directionality of the nuclear transport is determined by the RanGTP/GDP gradient across the nuclear envelope. Importins capture cargoes in the cytoplasm, release them upon encountering RanGTP inside nuclei, before returning to the cytoplasm for GTP-hydrolysis and another round of transport. Importin-7 (Imp7) is a member of the importin-β superfamily of NTRs that can act autonomously or in association with Importin-β1. Imp7 plays a role in the nuclear import of several proteins, including ribosomal proteins, histone H1, and the signaling effectors Smad3 and Erk[23–27].

A key signaling route controlling mechanotransduction pathways is led by the Hippo pathway core kinases mammalian STE20-like (MST) and large tumor suppressor kinase (LATS), which are regulated by a plethora of upstream signals and control the phosphorylation of YAP/TAZ, the Hippo pathway transcriptional regulators[28–30]. The Hippo pathway controls organ size and tissue homeostasis by regulating the expression of genes important in cell proliferation, apoptosis, and cell differentiation[28]. This gene expression regulation is initiated by the nuclear translocation of YAP/TAZ and followed by the binding and activation of TEAD and other transcription factors[28, 31–33]. YAP/TAZ have also been described to be part of gene expression circuits involving R-Smads (receptor-regulated Smads), including Smad2, 3 and 1, and Smad7[34–39], which results in both stimulatory[35, 39–42] and repressive[38, 43–45] gene expression. This reflects the complex and context-dependent interplay between the Hippo and TGFβ pathways (reviewed in[46]).

Multiple studies have shown that the nucleo-cytoplasmic balance of YAP/TAZ is tightly regulated by mechanical cues[18, 47–51]. Activation of MST1/2 leads to LATS1/2 phosphorylation, which in turn phosphorylates YAP on several serine residues, promoting YAP cytoplasmic retention[52, 53]. Mechanical cues positively regulate YAP/TAZ nuclear translocation by regulating the actomyosin system, which acts on or upstream of LATS1/2, and independent of LATS1/2[18, 47–50, 54, 55]. Tension-controlled signaling events downstream of YAP/TAZ are also important to regulate their nuclear accumulation[51]. However, the exact mechanism by which YAP is translocated into the nucleus in response to mechanical signals is still poorly understood. Based on its size (55 kDa), YAP should be imported into the nucleus assisted by an NTR, but its identity and the regulatory principles involved are unknown. Understanding this process is important, as alterations in the YAP/TAZ activity, which requires their import into nuclei, have been associated with cancer and other human diseases[52, 56–58].

Here, we identify a mechanoresponsive NTR, Imp7, that is indirectly regulated by mechanical cues, providing a mechanical control on the nuclear translocation process. This regulation is mediated by the Hippo pathway effector YAP, which functions as an active cargo of Imp7. Upon mechanical stimulation, YAP ensures its nuclear translocation by monopolizing Imp7, while restricting the association of other cargoes, such as Smad3 and Erk2. This allows the Hippo pathway to feed into other signaling pathways by regulating the nuclear import mediated by Imp7 in response to mechanical cues. Thus, the interaction between YAP and Imp7 ensures the nuclear translocation of YAP and provides a mechanism to tune different signaling pathways in response to mechanical cues.

## Results

**Imp7 is a mechanoresponsive NTR.** Cell confluence regulates several signaling pathways and affects the tensional status of cells[59]. To identify proteins sensitive to cell density-mediated nucleo-cytoplasmic shuttling, we performed an unbiased quantitative proteomics-based analysis in cells under different confluences. We sorted the proteins identified in the cytoplasm and the nucleus according to their change in relative amount between both compartments in low versus high-density conditions (Fig. 1a). Among the proteins whose relative amount in the nucleus increased with low density we identified an NTR (Importin-7, Imp7), three calcium regulators (Anxa2P2, Calr, and Anxa6), two proteins related to actin biology (Dstn and Capbz), several enzymes (Prdx1, Nme2, Pgam1, Dhfr, Gapdh, and Ldha) and other proteins (Hspe1 and H3-2). Among the proteins that decreased in the nucleus at low density, we identified several proteins related to transcription (Ddx17, Hnrnpcl2, Mybbp1A, and Ddx21) and one kinase (Stk39) (Fig. 1b, a complete list of identified proteins is shown in Supplementary Data 1). We reasoned that the nucleo-cytoplasmic transport machinery could be relevant to aid and regulate signals controlling mechanoresponsive nuclear proteins. For this reason, we validated and focused on Imp7, an NTR involved in the nuclear import of various proteins[22–24].

To confirm that Imp7 modified its nucleo-cytoplasmic balance upon changes in cell confluence, we stained for endogenous Imp7 in low and high cell confluency. The specificity of the Imp7 antibody for immunofluorescence was first validated by silencing Imp7 with two independent siRNAs (Supplementary Fig. 1a, b). In highly confluent cells, Imp7 was mostly localized in the cytosol, while in sparse cells Imp7 was enriched in the nucleus (Fig. 1c, d), in agreement with the quantitative proteomic analysis (Fig. 1b). The mass spectroscopy experiment identified several regulators of nucleo-cytoplasmic shuttling, but none of them showed a statistically significant change in their nucleo-cytoplasmic balance (Supplementary Table 1). To further confirm that the sensitivity to cell density was not shared by other proteins of the nucleo-cytoplasmic transport machinery, we stained for the well-characterized NTRs Importin-β1, Importin-α1, and Importin-α5, as well as for the GTPase Ran. We validated the specificity of the antibodies used for staining in transfected cells with the corresponding siRNAs (Supplementary Fig. 1c–j). Retinoblastoma (Rb) protein was used as a negative control to ensure that the sole change in cell morphology could not account for differences in the quantification (Fig. 1e, j). Contrary to Imp7, none of the other NTRs changed their nucleo-cytoplasmic ratio as a function of cell density (Fig. 1f–h, j). Quantification of the

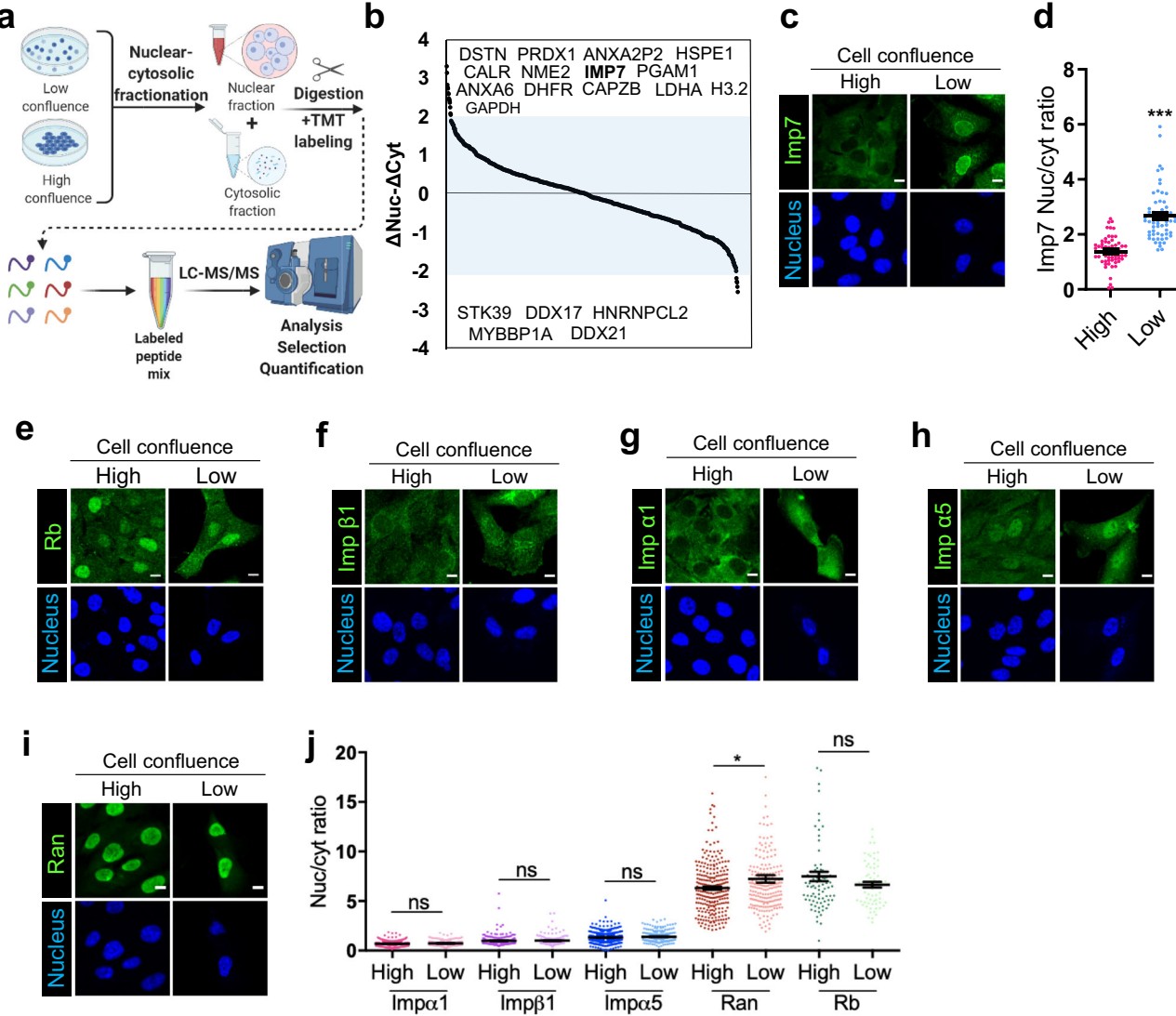

**Fig. 1 Imp7 nuclear accumulation is regulated by cell density. a** Schematic representation of the experimental design to identify proteins sensitive to changes in nucleo-cytoplasmic balance as a function of cell density by a quantitative MS approach. Created with Biorender.com. **b** Quantitative MS hypothesis-free analysis of nuclear and cytosolic fractions of RPE-1 cells grown at low (10,417 cells/cm$^2$) or high (218,750 cells/cm$^2$) confluence. For each protein identified, the difference of cytosolic change between low and high confluence was substracted to the difference of nuclear change between low and high confluence ($z$-score). We selected and sorted the 5% identified proteins with the higher and lower $z$-scores. **c, d** Immunofluorescence of endogenous Imp7 in RPE-1 cells plated at the high and low confluence. Quantification is shown in graph (**d**). $N = 58$ cells for each condition, from 3 independent experiments. $P$-value $= 1.100e-14$. **e-i** Immunofluorescence of the indicated endogenous proteins in RPE-1 cells plated at high and low cell confluence. Rb: retinoblastoma. Data representative from three biologically independent experiments. **j** Quantification of the nucleo-cytoplasmic intensity ratio of the indicated proteins in high and low-density cell cultures (high and low). From left to right in the graph, $N = 248$, 242, 222, 144, 401, 310, 266, 209, 80, and 63 cells per condition, from 3 independent experiments. Statistical analysis with a two-tailed unpaired $t$-test. From left to right, $P$-values $= 0.052$, 0.706, 0.067, 0.024, and 0.131. Raw data are available in the Source Data file. Data represent mean ± s.e.m. Scale bar 10 µm. $P$-values below or equal to 0.05, 0.01, or 0.005 were considered statistically significant and were labeled with 1, 2, or 3 asterisks, respectively.

nucleo-cytoplasmic distribution indicated that Ran showed a slight, but statistically significant, increase in its nuclear pool in low confluence condition versus high confluence condition (Fig. 1i, j), but, in comparison to Imp7, it was poorly regulated by cell density (~14% vs. ~93% increase in Imp7, Fig. 1d, j). This subtle change in Ran distribution could reflect the pool of Ran bound to Imp7. Thus, Imp7 is an NTR regulated by cell density that accumulates in the nucleus in sparse cells.

To further characterize Imp7 response to cell density, we used fibronectin-coated micropatterns of different sizes, which exclude the contribution of signaling initiated exclusively by cell–cell adhesion complexes[60] and allow comparing cells with low or high

spreading capacity (Fig. 2a). In small micropatterns (300 µm$^2$), where cell tension is lower and cells are not fully spread out[61, 62], Imp7 localized predominantly to the cytosol, while in large areas (2025 µm$^2$), where tension is higher[61, 62], Imp7 relocated predominantly to the nucleus (Fig. 2a–c), similar to fully spread cells (Fig. 1c). Next, we plated cells at low confluence but in different rigidity substrates, which impinge on tension-regulated pathways[61, 63]. In soft matrices with an elastic modulus of 2.3 kPa, mimicking soft tissues[64, 65], Imp7 was mainly localized to the cytosol, while in stiff substrates with an elastic modulus of 55 kPa, mimicking stiff tissues[61, 64, 65], it localized predominantly to the nucleus (Fig. 2d–f). To test whether similar results were observed

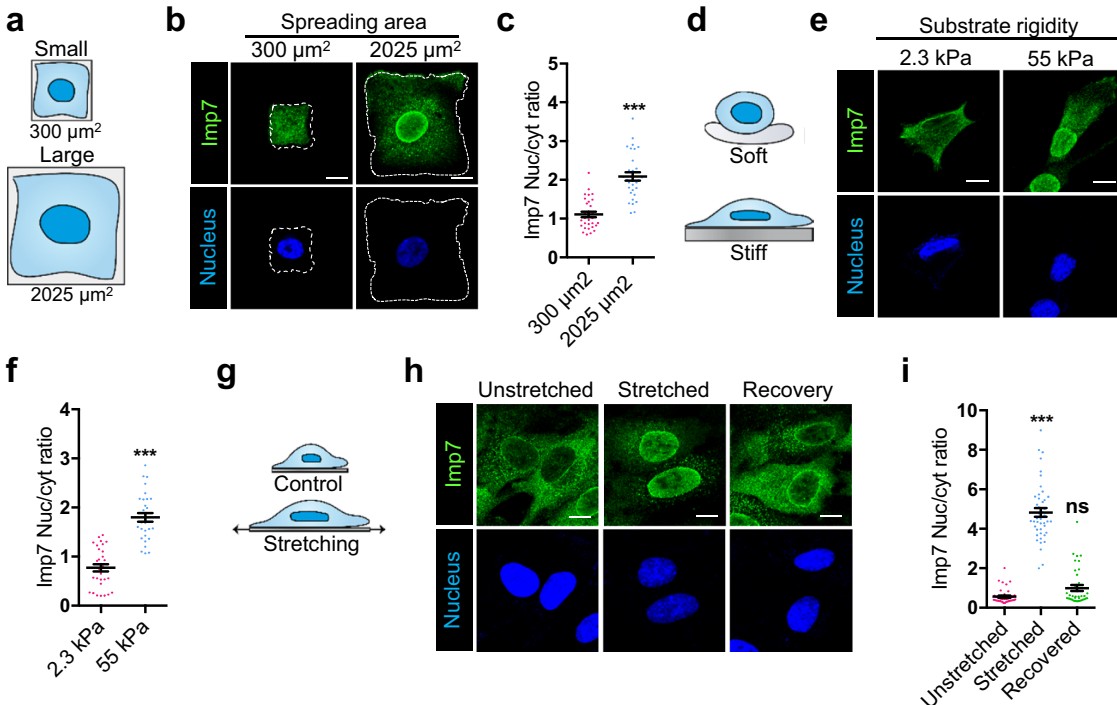

**Fig. 2 Imp7 nuclear accumulation is regulated by different mechanical cues. a** Diagram showing cell spreading in different area micropatterns used in (**b**, **c**). **b**, **c** Immunofluorescence of endogenous Imp7 and nuclei in RPE-1 cells plated on different area micropatterns (dashed line indicates cell boundary) (**b**). Quantification of the nucleo-cytoplasmic distribution of Imp7 is shown in (**c**). $N = 30$ cells per condition from 3 independent experiments. $P$-value = 1.435e−09. **d** Diagram showing cell spreading in different stiffness matrices used in (**e**, **f**). **e**, **f** Immunofluorescence of endogenous Imp7 and nuclei in RPE-1 cells plated on soft (2.3 kPa) and stiff (55 kPa) polyacrylamide matrices. Quantification of the nucleo-cytoplasmic distribution of Imp7 is shown in (**f**). $N = 30$ cells per condition from 3 independent experiments. $P$-value = 2.871e−12. **g** Diagram showing cell stretching used in (**h**, **i**). **h**, **i** Immunofluorescence of endogenous Imp7 and nuclei in MOVAS cells plated on silicone membranes and submitted to bi-axial stretching for 1 h and 20% amplitude. Recovering was obtained after 3 h. Quantification is shown in the right graph (**i**). $N = 60$ (control), 65 (stretching), and 54 (recovery) cells from 3 independent experiments. $P$-value = 1.252e−23 (stretched) and 0.061 (recovered). Raw data are available in the Source Data file. Statistical analysis with a two-tailed unpaired $t$-test. Data represent mean ± s.e.m. Scale bar 10 μm. $P$-values below or equal to 0.05, 0.01, or 0.005 were considered statistically significant and were labeled with 1, 2, or 3 asterisks, respectively.

in primary cells, we used mesenchymal stem cells (MSCs) obtained from human donors. In these cells, Imp7 was also redistributed to the nucleus in sparse cells while in highly confluent cells it was mostly cytosolic (Supplementary Fig. 2a, b). Similarly, in soft substrates, Imp7 was more cytoplasmic than in stiff substrates (Supplementary Fig. 2c, d).

All these tension regulating conditions were applied to cells over relatively long periods (>20 h). We sought to ascertain whether acute tension changes could also regulate Imp7 localization. Mechanical stretching of cells induces reinforcement of the actin cytoskeleton and increases tension in the cell[14, 66, 67]. Since muscle cells are physiologically exposed to stretching forces, we used aortic smooth muscle cells (MOVAS). As shown in Fig. 2g–i, Imp7 was located mostly to the cytosol in resting conditions, while upon mechanical stretching for 60 min it predominantly localized to the nucleus. Interestingly, when cells returned to non-stretched conditions, Imp7 distribution mimicked the original, pre-stretching situation. Collectively, these results suggest that Imp7 nuclear accumulation is highly sensitive to conditions generating increased cell tension.

**Actomyosin-controlled tension and Nesprin-1 regulate Imp7 nucleo-cytoplasmic balance.** In order to understand how Imp7 changes its nucleo-cytoplasmic distribution upon mechanical stimuli, we interfered with the actomyosin system that transmits forces sensed by integrins[68]. First, we disrupted the actin cytoskeleton with cytochalasin D (Cyt D) in epithelial cells and

primary MSCs. Cyt D treatment prevented Imp7 nuclear accumulation in sparse epithelial cells (Fig. 3a–c) and in primary MSCs (Supplementary Fig. 2e, f). Similarly, myosin II inhibition with blebbistatin, which prevents contraction and reduces actin-generated tension[63], reduced the nuclear pool of Imp7 (Fig. 3d, e). Furthermore, RhoA inhibition with Y27632, which prevents contraction and actin polymerization[69], also reduced the nuclear pool of Imp7 (Fig. 3f, g). Together, these results suggest that cell tension, controlled by the RhoA-actomyosin system, is important to regulate Imp7 nuclear distribution.

Tension reaching stress fibers can be transduced to the nucleus through the LINC (linker of the nucleoskeleton and cytoskeleton) complex[13, 70]. To determine whether this pathway was important for Imp7 distribution, we expressed the dominant-negative KASH domain of Nesprin-1, a component of the LINC complex connecting the SUN proteins to the actin cytoskeleton[71]. Cells expressing this fragment were stained for Imp7 and the nucleo-cytoplasmic ratio was determined. As shown in Fig. 3h, i, Imp7 distributed in the nucleus in sparse cells expressing GFP control while in cells expressing GFP-DN KASH Imp7 redistributed to the cytosol, suggesting that the connection of the actin cytoskeleton and the nucleoskeleton is important to regulate Imp7 distribution.

**YAP is actively imported into the nucleus by Imp7.** The sensitivity of Imp7 to mechanical cues faithfully recapitulates the behavior of YAP/TAZ transcriptional regulators[18, 51, 72]. Interestingly, Imp7

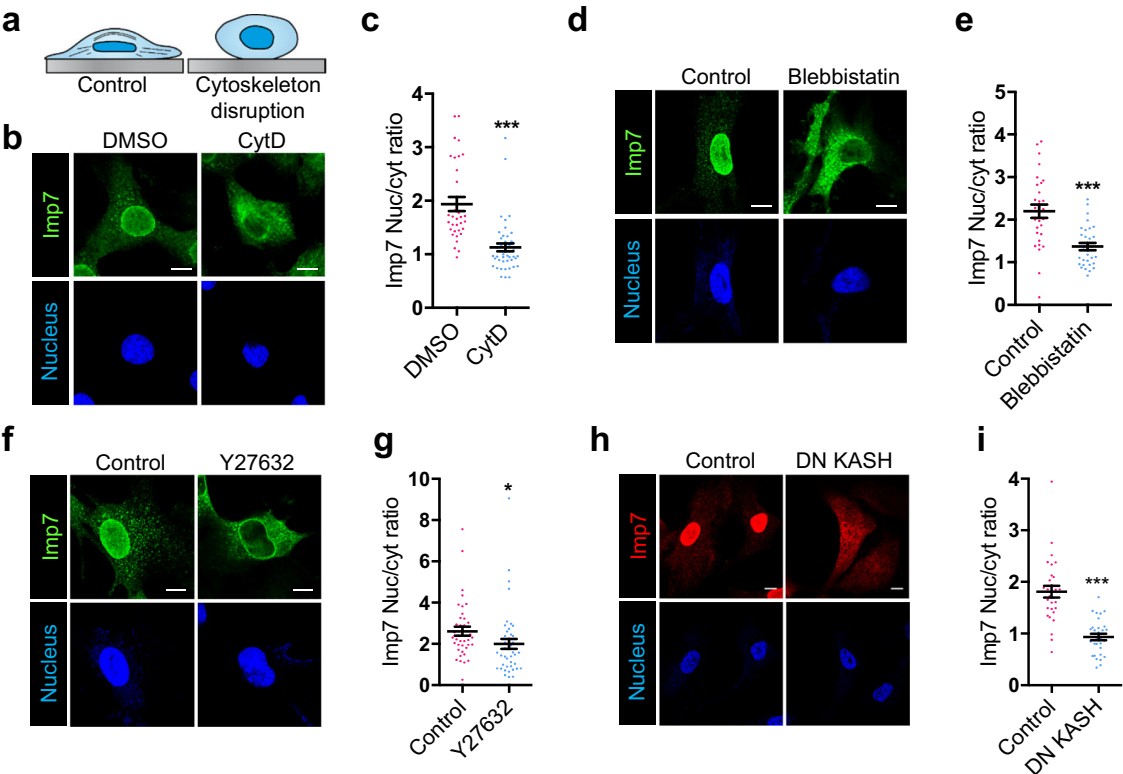

**Fig. 3 Actomyosin-controlled tension and Nesprin1 regulate Imp7 localization. a** Diagram showing loss of cell spreading upon disruption of the actin cytoskeleton. **b, c** Immunofluorescence of endogenous Imp7 and nuclei in RPE-1 cells upon 1 h of 1 µM Cyt D treatment (**b**). Quantification is shown in graph (**c**). $N = 33$ (control) and 47 (Cyt D treated condition) cells from 3 independent experiments. $P$-value $= 1.465e-06$. **d, e** Immunostaining for endogenous Imp7 (**d**) and quantification of its distribution (**e**) in RPE-1 cells, exposed to either vehicle or 10 µM blebbistatin for 1 h. $N = 30$ (control) and 31 (blebbistatin) cells from 3 independent experiments. $P$-value $= 2.641e-05$. **f, g** Immunostaining for endogenous Imp7 (**f**) and quantification of its distribution (**g**) in RPE-1 cells, exposed to either vehicle or 25 µM Y27632 (ROCK inhibitor) for 1 h. $N = 41$ (control) and 43 (treated) cells from 3 independent experiments. $P$-value $= 0.0413$. **h, i** Immunofluorescence of endogenous Imp7 and nuclei in RPE-1 cells upon overexpression of GFP (Control) or GFP-DN KASH (DN KASH) (**h**). Quantification is shown in graph (**i**). $N = 30$ and 31 cells, respectively, from a representative experiment of three independent experiments. $P$-value $= 2.751e-08$. Raw data are available in the Source Data file. Statistical analysis with a two-tailed unpaired $t$-test. Data represent mean ± s.e.m. Scale bar 10 µm. $P$-values below or equal to 0.05, 0.01, or 0.005 were considered statistically significant and were labeled with 1, 2, or 3 asterisks, respectively.

was identified, among other 89 potential interactors, in a previous YAP/TAZ interactome study[73]. Thus, we hypothesized that Imp7 could be a relevant node of mechanotransduction pathways and regulate YAP/TAZ nuclear import in response to mechanical cues. TAZ has been shown recently to enter the nucleus by a Ran-independent mechanism[74], which excludes Imp7 as a potential regulator of TAZ nuclear import, as Imp7 requires Ran[24]. Thus, we hypothesized that Imp7 could be responsible for YAP nuclear entry in response to mechanical cues.

First, to study the association of endogenous YAP with Imp7 in cells we performed proximity ligation assays (PLA), a method to detect in situ protein interactions with high specificity and sensitivity[75, 76]. As shown in Fig. 4a, staining of both endogenous YAP and Imp7 produced a strong signal, while any antibody alone or in combination with an unrelated antibody (Importin-α5 and MMP2) gave no signal (Fig. 4a). As expected, the PLA Imp7-YAP signal was mainly cytosolic, as both proteins would form a complex in this compartment prior to nuclear translocation and subsequent dissociation induced by Ran[77]. Importantly, knock-down of either Imp7 (Supplementary Fig. 1a) or YAP (Supplementary Fig. 3a-b) produced a significant reduction in the PLA signal (Supplementary Fig. 3c), validating the specificity of the assay. Thus, the association measured by PLA represents a specific Imp7-YAP association in cells between the endogenous proteins. As an independent test of association, we performed co-

immunoprecipitation assays on endogenous proteins. Immunoprecipitation of Imp7 efficiently co-immunoprecipitated endogenous YAP from cell lysates, while a control antibody did not (Fig. 4b). The Imp7-YAP interaction was performed using whole cell lysates, so the interaction could be mediated by another protein. To demonstrate that Imp7 and YAP interact directly, we purified proteins to perform GST pull-down assays (Supplementary Fig. 4a). GFP8Q-tagged YAP, but not control GFP8Q, interacted with pure GST-Imp7, but not with GST alone (Fig. 4c, lanes 1–3 and 6–8). Conversely, GST-YAP, but not GST alone, interacted with purified untagged Imp7 (Supplementary Fig. 4b). To determine the region in YAP needed for the interaction we focused on the last five amino acids of YAP, which are known to be required to enter the nucleus[78, 79]. As shown in Fig. 4c, YAP ΔC (lacking the C-terminal five amino acids) did not interact with Imp7. Thus, Imp7 and YAP form a complex in cells and in vitro that is dependent on the C-terminus of YAP.

To determine whether Imp7 was a bona fide NTR for YAP, we first mimicked nuclear pore perm-selectivity with FG hydrogels[80, 81]. This assay is based on the FG domain-controlled barrier at the nuclear pore complexes through which nucleo-cytoplasmic transport is conducted[20, 81, 82]. We recombinantly expressed and purified the scNup116 FG domain from *E. coli* and formed FG particles by phase separation. GFP8Q-YAP alone, in the absence of NTRs, stayed outside the hydrogels (with

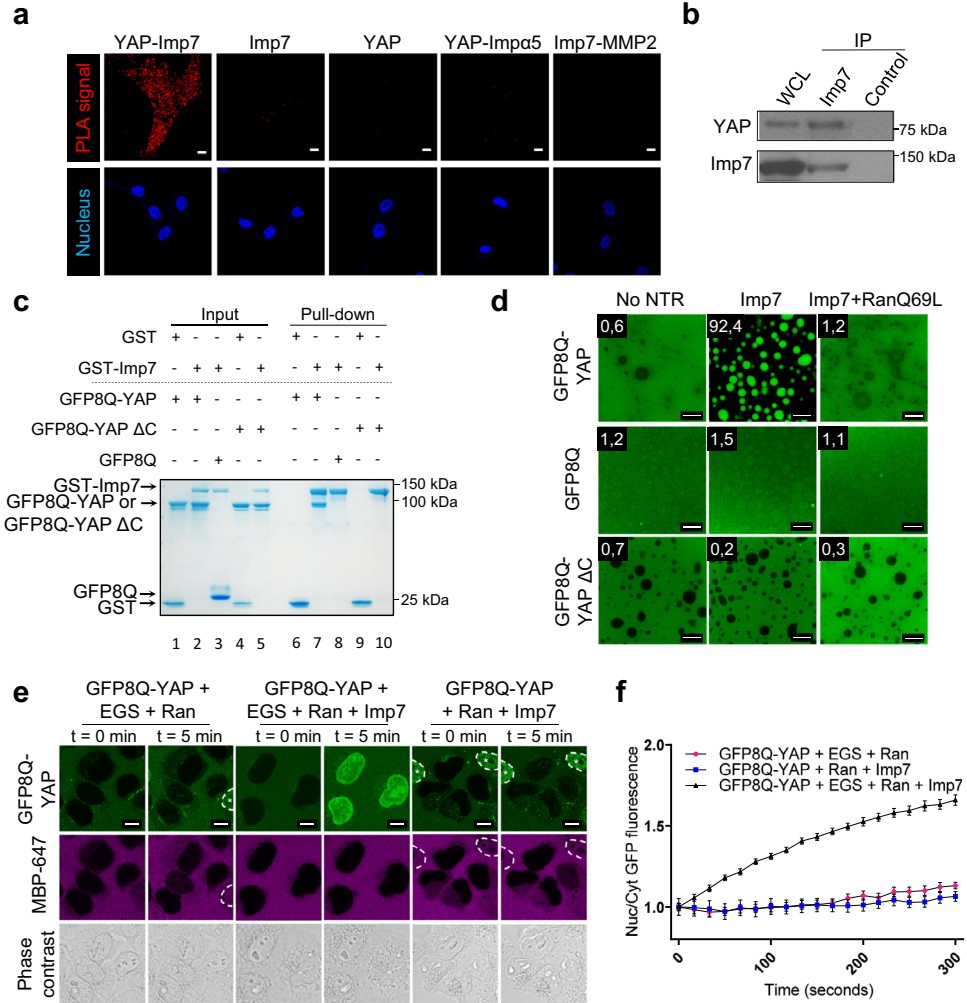

**Fig. 4 YAP is actively imported into the nucleus by the nuclear transport receptor Imp7. a** Association of endogenous Imp7 and YAP was analyzed by in situ PLA (red puncta) in RPE-1 cells using anti-YAP (63.7) and anti-Imp7 (rabbit) antibodies. Different controls are shown: anti-YAP or anti-Imp7 alone, and anti-YAP and anti-Imp7 with an unrelated antibody (anti-Impα5 and anti-MMP2, respectively). Images representative from three biologically independent experiments. **b** Lysates from RPE-1 cells grown at low confluence were immunoprecipitated with anti-Imp7 antibody. Immunoprecipitates and whole-cell lysates (WCL) were analyzed by immunoblotting with anti-YAP antibody and Imp7 as indicated in the figure. N-sMASE antibody was used as an immunoprecipitation antibody control. Three independent experiments were carried out. **c** Direct binding of YAP, but not YAP ΔC, to Imp7. GFP8Q, GFP8Q-YAP, or GFP8Q-YAP ΔC and GST-Imp7 or GST resin were combined (+) as indicated. Coomassie staining shows a fraction of the initial mixes (input) and pull-downs after the washes (right side). Representative of three biologically independent experiments. **d** Partitioning of GFP8Q-YAP, GFP8Q-YAP ΔC, and GFP8Q into FG particles derived from scNup116 FG domain. RanQ69L was used to block the interaction between cargo and the NTR. The brightness of image scans was individually adjusted. The average of partition coefficients is indicated for each condition. Representative of three independent experiments. **e**, **f** Import into nuclei of digitonin-permeabilized HeLa cells was followed for 5 min in the presence of an energy-regenerating system (EGS), a Ran mix, and 0.7 µM cargo GFP8Q-YAP that was pre-complexed with Imp7. Atto647N-labeled MBP (G260C mutant) was added as a control for the identification of leaky nuclei (cells labeled with white asterisks). Phase contrast was adjusted individually. Quantification is shown in graph (**f**). $N = 42$ cells in YAP + Imp7 + Ran, $N = 36$ cells in YAP + Ran, and $N = 40$ cells in YAP + Imp7 condition, data from 5 independent experiments. Raw data are available in the Source Data file. Statistical analysis with a two-tailed unpaired $t$-test. Data represent mean ± s.e.m. Scale bar 10 µm.

a partitioning coefficient (p.c.) of 0.6, while the addition of Imp7 induced a strong accumulation of GFP8Q-YAP inside the particles, with a p.c. of 92.4 (Fig. 4d). The addition of RanQ69L (a mutant locked in the RanGTP state) blocked GFP8Q-YAP influx into the FG phase (p.c. of 1.2), probably by prematurely releasing YAP from its importin (Fig. 4d). As expected, the GFP8Q tag alone did not enter regardless of the conditions used (maximum p.c. of 1.5). The addition of other NTRs to GFP8Q-YAP, including Trn1 (p.c. of 0.4), Imp5 (0.5), and Impβ1 (1.1) did not induce the translocation of YAP (Supplementary Fig. 4c). Interestingly, Imp8 slightly induced YAP translocation (p.c. of

5.8, Supplementary Fig. 4c). Imp7 and 8 share ~80% sequence similarity and other known cargoes of Imp7 are also imported by Imp8[26], which could explain this effect. Consistently, YAP ΔC, which did not bind Imp7 (Fig. 4c), was completely excluded from the FG particles (Fig. 4d), further supporting the specificity of the assay and the requirement of this region for Imp7-mediated nuclear import, consistent with the lack of nuclear accumulation of this mutant in cells[78, 79].

To further confirm the capability of Imp7 to import YAP, we used a cell-based nuclear import assay[82–84]. Permeabilized cells devoid of cytosolic factors were incubated with Ran and GFP8Q-

labeled YAP, and different combinations containing an energy generating system (EGS) and pure Imp7. Pure GFP8Q-YAP with the EGS and Ran was unable to enter the nucleus. In contrast, the addition of Imp7 effectively induced GFP8Q-YAP nuclear accumulation (Fig. 4e, f). As expected, Imp7 without the EGS was not sufficient to induce YAP nuclear import (Fig. 4e, f) and the GFP8Q tag did not enter the nucleus[81]. The residual signal observed in some cells under some conditions (marked and labeled with an asterisk) was due to unspecific nuclear leakiness, as the inert fluorophore-labeled protein control (in magenta) also entered the nucleus. Thus, Imp7 is sufficient to drive YAP nuclear import in vitro.

**Imp7 is required for YAP/TAZ nuclear accumulation in cells.** Next, we determined whether Imp7 was important for YAP nuclear localization in living cells. To detect endogenous YAP by immunofluorescence we used a broadly used YAP antibody[18, 51, 73] (Santa Cruz 63.7). However, this antibody also recognizes TAZ to some extent[18], thus we refer to both YAP and TAZ in the following assays. First, we silenced Imp7 with two independent siRNA sequences in epithelial cells (Fig. 5a, quantification in Supplementary Fig. 5a) and stained for YAP/TAZ. Imp7 depletion inhibited the nuclear accumulation of YAP/TAZ in epithelial cells, as compared to control cells (Fig. 5b, c). Neither YAP protein/mRNA levels nor the extent of LATS1/2-induced YAP serine 127 phosphorylation (pS127 YAP)[52] was affected by Imp7 depletion (Fig. 5a, quantified in Supplementary Fig. 5a–d). When two other antibodies (CST 14074 and Santa Cruz H-9) reacting mainly against YAP were used (previously validated[18] and confirmed in Supplementary Fig. 5e), a similar reduction in the nucleo-cytoplasmic ratio was observed when Imp7 was silenced, further confirming the previous results (Supplementary Fig. 5f, g). Rescue of Imp7 expression using a siRNA-insensitive *IPO7* (Imp7) coding sequence (from *Xenopus laevis*) in Imp7 silenced cells partially restored nuclear YAP/TAZ, suggesting that the effect of Imp7 depletion on YAP/TAZ localization was specific (Fig. 5d, e). Depletion of Imp7 in primary MSCs also prevented YAP/TAZ nuclear accumulation, independently on the antibody used to stain YAP (Supplementary Fig. 5h–j) without affecting the total levels of YAP, nor pS127 YAP (Supplementary Fig. 5k, l). Thus, Imp7 is necessary for YAP nuclear localization in cells.

Some studies have suggested functional and/or biochemical connections between YAP/TAZ and other importins[85, 86]. For this reason, we independently silenced all α and β importins (Supplementary Fig. 5m, a total of 19) and determined their effect in the nucleo-cytoplasmic balance of YAP/TAZ. As shown in Fig. 5f, most of the importins were irrelevant for YAP/TAZ nuclear accumulation, except for Importin α1, α5 and α8, and Imp7. Importin α1, α5, and α8 had a mild effect on nuclear YAP/TAZ (about 15% reduction); this effect was much weaker than the one obtained upon Imp7 silencing (about 70% reduction), suggesting that Imp7 is the main importin responsible for YAP/TAZ nuclear accumulation (Fig. 5f). Double knockdown of Importin α1 and β1, or Imp7 and its highly homologous Imp8 did not synergize (Fig. 5f). To determine whether these importins were acting in parallel, independent pathways, we silenced them in different combinations. Simultaneous silencing of Imp7 and each of the α importins affecting YAP/TAZ localization (Importin α1, α5, α8), the triple knockdown of Importin α1, α5, and α8, and the quadruple knockdown of Importin α1, α5, α8, and Imp7 did not improve the effect obtained by silencing only Imp7 (Fig. 5g), suggesting that Importin α1, α5, and α8 are not acting in parallel routes to Imp7, but rather in a linear pathway. The knockdown efficiency was equal when an individual or multiple siRNAs were used (Supplementary Fig. 5n), ruling out inefficient knockdowns

when multiple siRNAs were used. Taken together, these results suggest that Imp7 is the main importin responsible for YAP/TAZ nuclear accumulation.

**Imp7 is necessary for YAP/TAZ-dependent gene expression.** YAP/TAZ positively regulate specific gene expression signatures through interaction with TEAD transcription factors[31–33, 87]. To test whether Imp7 functionally regulated YAP/TAZ, we first monitored the impact of Imp7 depletion on the TEAD transcriptional activity based on the TEAD-driven 8× GTIIC luciferase reporter[18]. Imp7 depletion significantly reduced the activity of the luciferase reporter (Fig. 5h). YAP transcriptional function is dependent on substrate rigidity, cell density, and mechanical stretching[18, 52, 72]. Thus, we analyzed the expression of YAP/TAZ target genes *Ctgf*, *Cyr61*, and *Ankrd1*[73] in cells treated under different conditions that regulate the cell tensional status, i.e., cell confluence (Fig. 5i), substrate rigidity (Fig. 5j), and mechanical stretching (Fig. 5k). In low confluence, stiff substrate, and under mechanical stretch, YAP/TAZ target genes were induced, as previously described[18, 72, 73]. Importantly, both basal expression levels and induction for all genes tested were significantly reduced upon Imp7 depletion across conditions (Fig. 5i–k). Considering that TAZ is imported into the nucleus in a Ran-independent manner and therefore independent of Imp7[24, 74] and that Imp7 imports pure YAP in vitro and in cells (Figs. 4 and 5), these results suggest that Imp7 is responsible for YAP function in mammalian cells.

**Msk is necessary for Yki induced organ growth in vivo.** We used *Drosophila* as an animal model to specifically address whether Imp7 was required for YAP function in vivo. In *Drosophila*, Imp7 is encoded by its homolog Moleskin (Msk)[27], while Yki is the homolog of YAP/TAZ[88]. Yki overexpression in *Drosophila* leads to tissue overgrowth[88]. Thus, we used this model to determine whether Msk was necessary for Yki-driven tissue overgrowth in vivo. To analyze this, we first overexpressed Yki (UAS-Yki) in the dorsal compartment of the wing imaginal disc of *Drosophila* larvae (apterus-Gal4), which resulted in tissue overgrowth compared to control larvae (Fig. 6a, b), as previously published[88]. Importantly, silencing of Msk (Supplementary Fig. 6a, b) in *Yki*-overexpressing discs fully rescued the normal size of the dorsal compartment (Fig. 6a, b), suggesting that Msk is needed for Yki-dependent organ growth in vivo. To test whether in this tissue Msk was regulating Yki nuclear accumulation, we stained for Yki in control and *Msk* knockdown flies. In these flies, Yki was localized to the cytoplasm in some cells, while in others it localized throughout the cytosol and the nucleus (Fig. 6c), consistent with other studies[89–91] and the observed tissue overgrowth phenotype (Fig. 6a, b). In contrast, upon *Msk* depletion, Yki accumulated in the cytosol in virtually all cells (Fig. 6c, quantified in d). To confirm this result, we overexpressed, in imaginal discs, GFP-tagged *Yki*, which was mostly cytoplasmic in the majority of cells but localized throughout the cytosol and the nucleus in some cells (Supplementary Fig. 6c). In contrast, upon *Msk* depletion, all cells showed cytoplasmic Yki-GFP localization (Supplementary Fig. 6c, quantified in d). Thus, *Msk* depletion prevents Yki nuclear accumulation in vivo. Next, we asked whether Msk subcellular localization was similar to Yki, as Imp7 was to YAP/TAZ (Figs. 1 and 2). The staining of endogenous Msk in wing imaginal discs showed that it had mainly two distinct patterns, cytoplasmic or cytoplasmic and nuclear, and these patterns were mirrored by Yki on a cell by cell basis (Fig. 6e). As a consequence, the correlation of both stainings was high (Fig. 6e merge panels, Pearson´s coefficient of 0.85 ± 0.04 SD). Similar results were obtained in

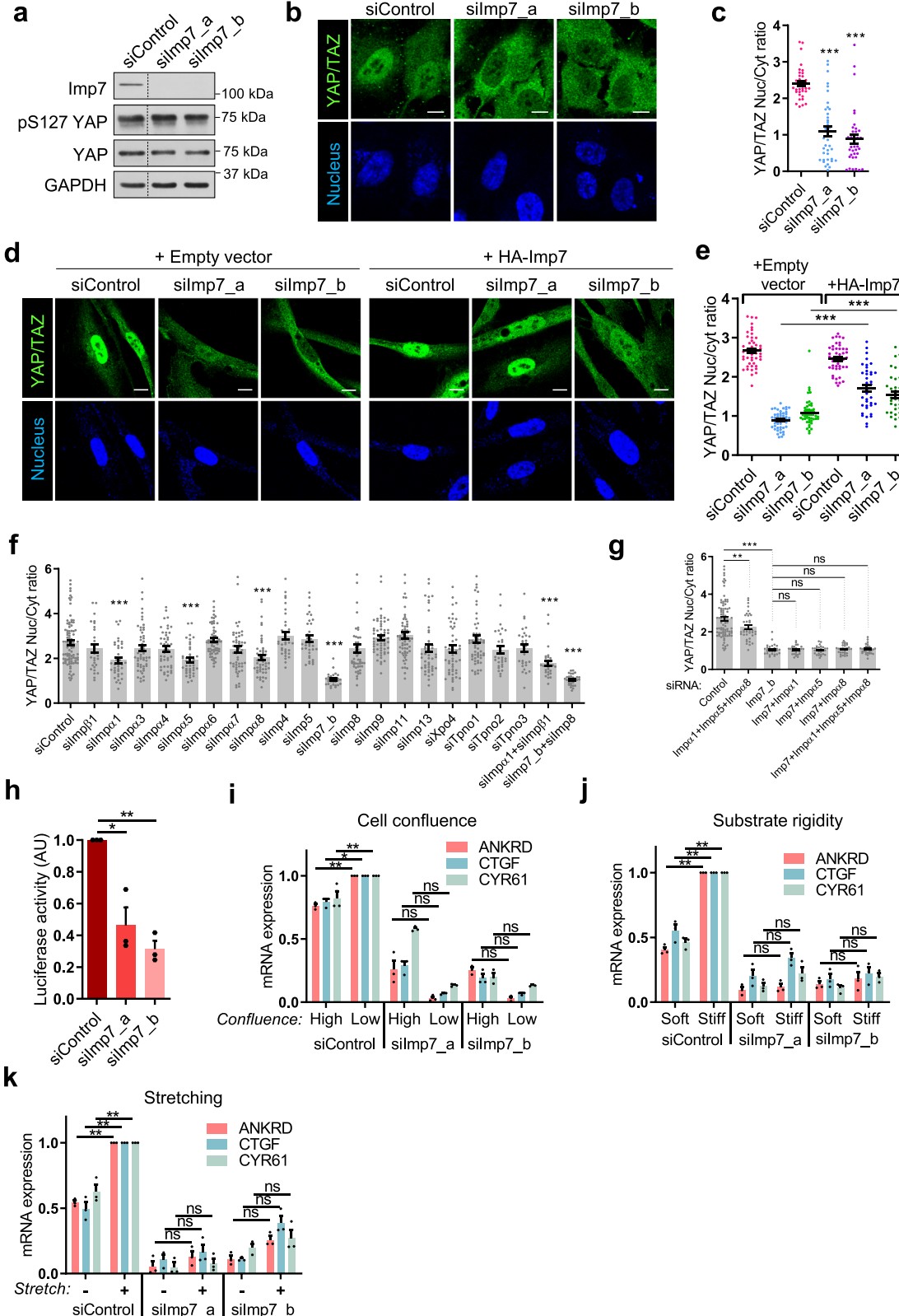

transgenic flies expressing YFP-Msk[92], where the YFP-Msk signal and the staining with anti-Msk showed a similar pattern to endogenous Yki (Fig. 6f, Pearson's coefficient of 0.912 ± 0.035 SD). To gain more insight into the mechanism that regulates Msk subcellular localization, we used *Drosophila* S2 cells. In plastic dishes, these cells grow in suspension and semi-adherent

conditions, but when plated on concanavalin A (ConA)-coated surfaces they fully spread out[93]. In uncoated dishes, Msk was mostly cytoplasmic, like Yki (Supplementary Fig. 6e, f). However, induction of cell spreading on ConA-coated surfaces induced Msk nuclear accumulation, which was mirrored by Yki (Supplementary Fig. 6e, quantified in f). Interestingly, treatments or

**Fig. 5 Imp7 is required for YAP/TAZ nuclear accumulation in cells, and YAP function in gene expression. a** Immunoblot showing specific depletion of Imp7 with two independent siRNAs. GAPDH is the loading control. YAP and pS127 YAP blots were performed in different gels simultaneously. Quantification is shown in Supplementary Fig. 5a–d. Irrelevant lanes in the blot were removed and denoted by the dotted line. **b, c** Immunofluorescence (**b**) and quantification (**c**) of the localization of endogenous YAP/TAZ in RPE-1 cells transfected with either control or Imp7 siRNAs. $N = 38$ cells per condition from three independent experiments. $P$-value $= 1.438e-11$ (siImp7_a) and $1.709e-15$ (siImp7_b). **d, e** Immunofluorescence (**d**) and quantification (**e**) of endogenous YAP/TAZ localization and nuclei in RPE-1 cells silenced with two independent Imp7 siRNAs. An empty vector or a *X. laevis* sequence of HA-Imp7, siRNA-insensitive, were overexpressed (**d**). Starting from the right, $N = 48, 48, 48, 47, 38,$ and 35 cells per condition, from 3 independent experiments. $P$-value $= 9.679e-11$ (siImp7_a) and $1.928e-05$ (siImp7_b). **f, g** Quantification of the effect on YAP/TAZ nucleo-cytoplasmic balance upon silencing of the indicated importins. $N$ range $= 30$ to 37 cells per condition from 3 independent experiments. $P$-values from left to right: panel **f**, 0.229, $1.149e-05$, 0.178, 0.093, $1.742e-06$, 0.309, 0.114, 0.0001, 0.106, 0.316, $4.448e-26$, 0.262, 0.106, 0.059, 0.287, 0.176, 0.353, 0.126, 0.261, $1.969e-07$, and $1.790e-26$; panel **g**, 0.007, $4.641e-26$, 0.536, 0.525, 0.322, and 0.339. **h** TEAD transcriptional activity in RPE-1 cells expressing the 8× GTIIC-luciferase reporter. Cells were silenced with control or Imp7 siRNAs. Luciferase activity was analyzed as described in Methods. Data are normalized to control siRNA silenced cells. Three independent experiments were analyzed. $P$-value $= 0.041$ (siImp7_a) and 0.005 (siImp7_b). **i–k** qRT-PCR analysis of *Ankirin1 (ANKRD), Ctgf* and *Cyr61* expression in RPE-1 cells transfected with either control or Imp7 siRNAs, and cultured at low/high confluence for 24 h (**i**), on stiff (55 kPa) or soft (2.3 kPa) substrates for 24 h (**j**), or under non-stretching/stretching conditions (biaxial stretching, 1 h) (**k**). Data were normalized to low confluence (**i**), stiff substrate (**j**), and stretching (**k**). Data from three biological replicates for each experiment. Statistical analysis with a two-tailed unpaired *t* test. Data represent mean ± s.e.m. $P$-values from left to right **i**: 0.005, 0.012, 0.009, 0.078, 0.084, 0.059, 0.074, 0.057, and 0.288; **j**: 0.006, 0.007, 0.009, 0.559, 0.092, 0.154, 0.548, 0.519, and 0.061; **k**: 0.0062, 0.0099, 0.0061, 0.263, 0.448, 0.619, 0.063, 0.071, and 0.393. Scale bar 10 μm. Raw data available in the Source Data file. $P$-values below or equal to 0.05, 0.01, or 0.005 were considered statistically significant and were labeled with 1, 2, or 3 asterisks, respectively.

conditions that reduced cell tension, such as actin cytoskeleton disruption with cytochalasin D (Supplementary Fig. 6g, h) or high cell density (Supplementary Fig. 6i, j), prevented Msk nuclear accumulation in ConA-coated dishes. Thus, conditions that increase tension favor Msk nuclear accumulation, as occurs with Yki[47–49, 54, 91, 94].

Next, we tested whether Msk interacted with Yki. In spread S2 cells plated on ConA surfaces, Msk interacted with Yki (Fig. 6g, h). Similarly, the constitutively active Yki S168A mutant[90] also interacted with Msk in spread S2 cells (Fig. 6g, h). Moreover, the interaction Msk-Yki was fully dependent on the C-terminus of Yki, as deletion of the last 7 amino acids (Yki ΔC) prevented binding to Msk (Fig. 6g, h). Interestingly, Yki and Msk did not interact in semi-detached conditions (Supplementary Fig. 6k, l), consistent with their inability to accumulate in the nucleus (Supplementary Fig. 6e, f). In contrast, constitutively active Yki S168A mutant, which bypasses the inhibitory effect of Warts[53, 90] in low tension conditions[47–49, 54, 94], interacted with Msk also in semi-detached conditions (Supplementary Fig. 6k, l). Collectively, these results suggest that Msk binds Yki under conditions that favor high cell tension, regulates its nuclear accumulation, and is required for Yki-induced organ growth in vivo. In addition, Msk localization, like in the case of Imp7, is sensitive to conditions regulating cell tension.

**Hippo pathway kinases regulate the Imp7–YAP interaction downstream of mechanical cues.** To understand the mechanistic details governing the Imp7–YAP interaction, we interfered with the tension generating machinery and the Hippo pathway. The sensitivity of Imp7 (Figs. 1–3) and YAP[18] to actin cytoskeleton-regulated tensional status of the cell raised the possibility that their interaction could be regulated by these cues. To test this, we first immunoprecipitated YAP from control cells or cells treated with Cyt D to determine whether the actin cytoskeleton was regulating the interaction. In sparse cells treated with vehicle, the interaction was efficiently detected. However, in Cyt D treated cells the interaction was completely abolished (Fig. 7a). In order to establish whether other conditions regulating cell tension could also regulate the Imp7–YAP interaction, we immunoprecipitated YAP from cells under different confluences. In sparse cells both proteins interacted, as YAP efficiently immunoprecipitated Imp7, while a control antibody did not (Fig. 7b, lanes 1, 3, 5). In contrast, in highly confluent cells, YAP did not interact with Imp7, as

it was not detected in the immunopurified fractions, similar to control antibody immunoprecipitates (Fig. 7b, lanes 2, 4, 6). To further understand the pathways regulating the interaction we used PLA targeted to Imp7 and YAP. In this setting, Cyt D treatment completely prevented the generation of the signal (Fig. 7c, d), consistent with the lack of interaction observed by co-immunoprecipitation (Fig. 7a). The addition of blebbistatin or Y27632 completely abolished the signal, suggesting that myosin II and RhoA were necessary for the interaction to occur (Fig. 7c, d). Collectively, these results suggest that cell confluence and the RhoA-actomyosin pathway regulate the binding of Imp7 to its cargo YAP.

In order to elucidate how actin-generated tension was regulating the interaction between Imp7 and YAP, we tested different conditions (actin polymerization disruption and inhibition of the Hippo pathway kinase MST1/2) across cell confluences. The inability of YAP to interact with Imp7 in high confluence was fully rescued by the inhibition of MST1/2 with XMU-MP-1[95] (Fig. 7e, f), suggesting that MST1/2 or upstream factors are activated by high cell density, preventing binding of YAP and Imp7. Similarly, Cyt D-induced disruption of the interaction in sparse cells was fully rescued by MST1/2 inhibition (Fig. 7g, h). These observations suggest that the actin cytoskeleton and low cell density inhibit MST1/2 or upstream of it, favoring the formation of the complex. MST1/2 kinases activate LATS1/2 kinases, leading to YAP phosphorylation and cytoplasmic retention[28]. To test whether LATS1/2-induced YAP phosphorylation was regulating the interaction with Imp7, we mutated serine 127 (a phosphoacceptor in YAP sequence for LATS1/2[52]) to glutamic acid to create a phosphomimetic mutant. As control of specificity, we mutated serine 128 to glutamic acid (S128E). YAP S127E mutant, similar to YAP ΔC, and contrary to YAP S128E, barely interacted with Imp7 in vitro (Fig. 7i, j, purified proteins are shown in Supplementary Fig. 4a). In cells, similar results were obtained, wild-type YAP and YAP S128E interacted with Imp7, while YAP ΔC and YAP S127E did not (Supplementary Fig. 7a, b, d). As expected, constitutively active and non-phosphorylatable YAP S5A mutant interacted like the wild-type form (Supplementary Fig. 7a, d). Collectively, these observations suggest that the specific phosphorylation on serine 127, regulated by the Hippo pathway, prevents the formation of the Imp7–YAP complex. To gain more insight into the regions of YAP relevant to interact with Imp7, we mutated three independent regions

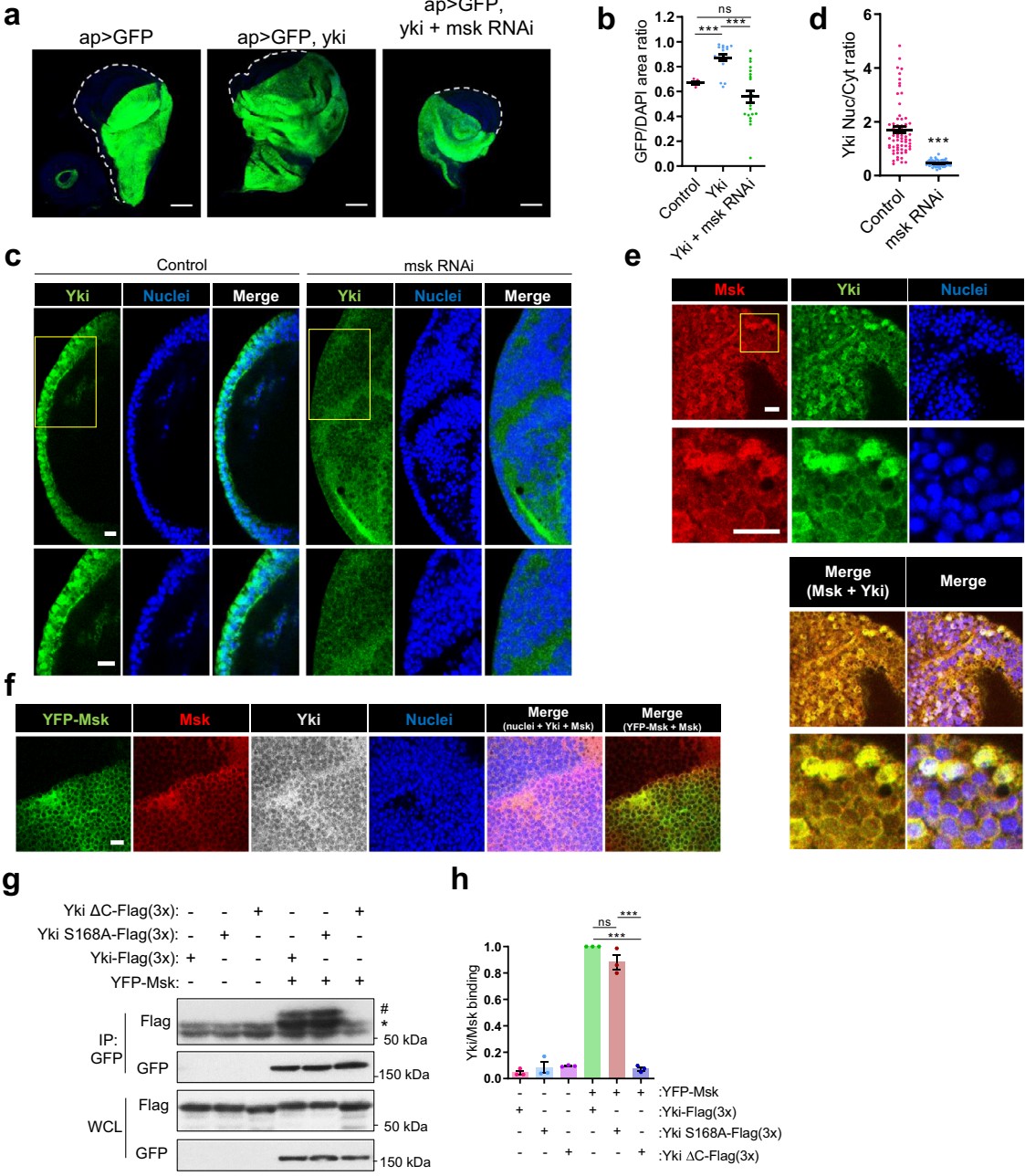

**Fig. 6 Msk is required for Yki nuclear accumulation and Yki-induced organ growth in vivo. a**, **b** Confocal images of *Drosophila* imaginal discs overexpressing GFP, GFP-*Yki*, and GFP-*Yki* + *msk* siRNA under the UAS/Gal4 system in the dorsal compartment (under the control of apterusGal4). Quantification in the graph (**b**). From left to right, *N* = 7, 18, and 22 discs. Scale bar 100 µm. *P*-value = 2.870e−07 (column 1 vs. 2), 2.392e−06 (column 2 vs. 3) and 0.084 (column 1 vs. 3). **c**, **d** Confocal images of *Drosophila* imaginal discs overexpressing *Yki* and *Yki* + *msk* siRNA under the UAS/Gal4 system in the dorsal compartment. Quantification of Yki nucleo-cytoplasmic ratio is shown in graph (**d**). Zooms are shown below. *N* = 66 and 54 cells from a representative experiment of 3 independent experiments. Scale bar 10 µm. *P*-value = 1.007e−14. **e** Confocal images of *Drosophila* imaginal discs overexpressing *Yki* and stained against Yki and Msk. Zooms are shown below. Data representative from 3 biologically independent experiments. Scale bar 10 µm. **f** Confocal images of *Drosophila* imaginal discs overexpressing *YFP-msk* under the UAS/Gal4 system in the dorsal compartment (under the control of apterusGal4) and stained against Yki and Msk. Data representative of three biologically independent experiments. Scale bar 10 µm. **g**, **h** S2 cells overexpressing different combinations of the indicated proteins were grown on top of a ConA coated surface. YFP-Msk was immunoprecipitated with anti-GFP. The immunopurified complexes and total cell lysates (WCL) were immunoblotted as indicated. IgG was marked with an asterisk and Yki was marked with a #. Quantification from three biologically independent co-immunoprecipitations is shown (**h**). Statistical analysis with a two-tailed unpaired *t*-test. Data represent mean ± s.e.m. ns non-significant. *P*-value = 0.167 (4th vs. 5th bar), 0.00023 (4th vs. 6th bar) and 0.0029 (5th vs. 6th bar). Raw data are available in the Source Data file. *P*-values below or equal to 0.05, 0.01, or 0.005 were considered statistically significant and were labeled with 1, 2, or 3 asterisks, respectively.

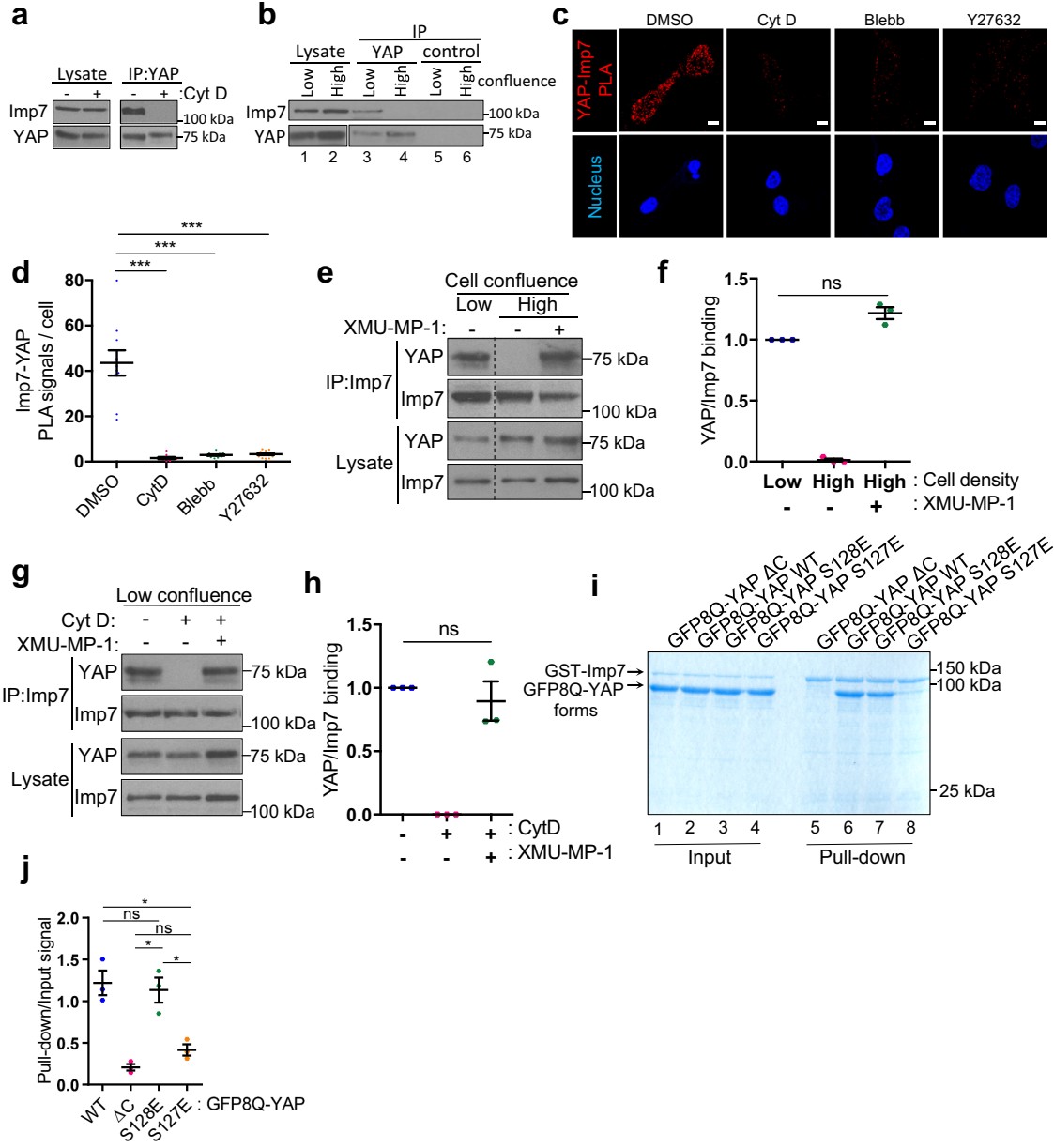

**Fig. 7 Cell density, actomyosin-controlled tension, and the Hippo pathway regulate Imp7–YAP complex formation. a** Cells at low confluency were treated with 1 μM Cyt D for 1 h or control vehicle, lysates were made, and YAP was immunoprecipitated. The immunopurified complexes and cell lysates were immunoblotted as indicated. Representative of three biologically independent experiments. **b** RPE-1 cells were grown to either low (10,417 cells/cm², cultured for two days) or high confluence (67,708 cells/cm²). Anti-YAP or control IgG (anti-ESE-1) were used to immunopurify associated complexes and were immunoblotted as indicated. Representative of three biologically independent experiments. **c, d** In situ PLA detection of the association between endogenous YAP (63.7) and Imp7 in RPE-1 cells treated with DMSO, Cyt D, blebbistatin, or Y27632. **d** Quantification of the PLA signal. From left to right, $N = 10, 9, 12$, and 12 fields per condition, from 3 independent experiments. P-value = 3.506e−05 (Control vs. Cyt D), 4.679e−05 (Control vs. Blebb) and 4.903e−05 (Control vs. Y27632). **e, f** Cells were grown at low or high confluency. High confluency cells were treated with vehicle (−) or MST1/2 inhibitor (+, XMU-MP-1) at 6 μM for 16 h. Cells were lysed and Imp7 was immunoprecipitated. Immunoprecipitated fractions and whole-cell lysates were blotted as indicated. Quantification of 3 biologically independent experiments is shown (**f**). Irrelevant lanes in the blot were removed and denoted by the dotted line. **g, h** Cells at low confluency were treated with vehicle (−) or MST1/2 inhibitor (+, XMU-MP-1) at 6 μM for 16 h. One hour before lysis, cells were (+) or were not (−) treated with Cyt D at 1 μM. Cells were lysed and Imp7 was immunoprecipitated. Immunoprecipitated fractions and lysates were blotted with the indicated antibodies. Quantification of three biologically independent experiments is shown (**h**). **i, j** Direct binding of YAP WT and YAP S128E, but not YAP ΔC or S127E, to GST-Imp7. GFP8Q-YAP WT, GFP8Q-YAP S128E, GFP8Q-YAP S127E, or GFP8Q-YAP ΔC and GST-Imp7 resin were combined. Coomassie staining shows a fraction of the initial mixes (input) and pull-downs. Quantification of the GFP8Q-YAP Coomassie bands are shown (**j**). $N = 3$ biologically independent experiments. Statistical analysis with a two-tailed unpaired t-test. Data represent mean ± s.e.m. Scale bar 10 μm. ns non-significant. P-value = 0.018 (WT vs. S127E), 0.703 (WT vs. S128E), 0.070 (ΔC vs. S127E), 0.019 (ΔC vs. S128E), and 0.025 (S127E vs. S128E). Raw data are available in the Source Data file. P-values below or equal to 0.05, 0.01, or 0.005 were considered statistically significant and were labeled with 1, 2, or 3 asterisks, respectively.

(represented in Supplementary Fig. 7e): (i) basic residues R88, K89, R203, and K204 were mutated to A (named ΔRK[2]); (ii) a second basic region (amino acids 314–320 "EKERLRL") resembling a recently proposed potential NLS for Imp7[96], was mutated to NAAIRSA to minimize changes in protein structure[97] (this mutant was named ΔNLS7); and (iii) three independent regions, resembling another proposed potential NLS for Imp7[98], were mutated to AAA: positions 143–145 (TPT), 154–156 (TPT) and 412–414 (TPD) (this mutant was named ΔTPT). Co-immunoprecipitation of endogenous Imp7 with the different YAP mutants showed that EYFP-YAP ΔNLS7 (Supplementary Fig. 7c, lane 4) did not co-immunoprecipitate with Imp7, while EYFP-YAP ΔRK[2] and EYFP-YAP ΔTPT co-immunoprecipitated with Imp7, although to a lesser extent (Supplementary Fig. 7c, d). Localization of these mutants indicated that EYFP-YAP ΔNLS7 barely accumulated in the nucleus, like EYFP-YAP ΔC[78, 79], while EYFP-YAP ΔRK[2] and EYFP-YAP ΔTPT were able to accumulate, although to a lesser extent in the case of EYFP-YAP ΔRK[2] (Supplementary Fig. 7f, g). In summary, YAP requires regions 500–504 and 314–320, and unphosphorylated serine 127[52] to interact with Imp7 and to accumulate in the nucleus (Supplementary Fig. 7d–g).

**YAP controls Imp7 mechano-responsiveness and binding to other cargoes.** While the role of the YAP-Imp7 complex on YAP biology was evident (Figs. 4–6), it was unclear whether this complex had any functional effect on Imp7 biology. Given the remarkable similarity of the mechanoresponse of YAP[18, 51, 72] and Imp7 (Figs. 1–3), we hypothesized that the Imp7–YAP complex could regulate the Imp7 mechanoresponsive pattern. Thus, we studied the nucleo-cytoplasmic distribution of Imp7 under high tension conditions in cells silenced for YAP. Unexpectedly, silencing of YAP with two independent siRNAs led to a strong reduction of Imp7 nuclear localization in cells growing at low density (Fig. 8a, b), while it did not affect total Imp7 levels (Supplementary Fig. 3a, d). This change in Imp7 localization was similar to the effect caused by Cyt D (Fig. 3b, c). The combination of Cyt D and YAP knockdown did not produce a synergistic effect, suggesting that YAP and the actin cytoskeleton were part of the same pathway regulating Imp7 sensitivity to tension (Fig. 8a, b). To further demonstrate that YAP was regulating Imp7 mechanoresponse, we prevented YAP phosphorylation by MST1/2 inhibition in highly confluent cells, where Imp7, like YAP, was mostly cytosolic (Fig. 1c, d). Interestingly, MST1/2 inhibition, which restored the formation of the Imp7–YAP complex (Fig. 7e), led to Imp7 nuclear accumulation in highly confluent cells, suggesting that the Hippo pathway regulates Imp7 nucleo-cytoplasmic balance (Fig. 8c, d). These results suggest that cargo binding can determine the nucleo-cytoplasmic balance of Imp7. To test whether other Imp7 cargo could also have a similar effect, we silenced Smad3, a cargo of Imp7[25, 26]. First, we confirmed that Imp7 silencing resulted in less Smad3 nuclear accumulation (Supplementary Fig. 8a, b), consistent with its role in Smad3 nuclear import[25, 26]. As shown in Fig. 8e, f, effective Smad3 silencing (Fig. 8g) did not alter the nucleo-cytoplasmic ratio of Imp7, neither in basal conditions nor upon Smad3 activation by TGFβ stimulation, which induces its nuclear translocation[99]. Thus, the effect of YAP on Imp7 nucleo-cytoplasmic balance was specific and not shared by another cargo, such as Smad3.

The Hippo pathway crosstalks with many other signaling pathways[28, 100–102] and some of these pathway's effectors use also Imp7 to translocate into the nucleus[25, 27, 103, 104]. Therefore, the strong effect of YAP on Imp7 localization (Fig. 8) could have an effect on those other effectors requiring Imp7 to shuttle into the nucleus. Thus, the Imp7–YAP complex formation would not only

drive YAP nuclear import but tune other pathways through Imp7. To test this hypothesis, we initially focused on Smad3, as multiple studies have shown crosstalk between the TGFβ and the Hippo pathways[46], and its nuclear import in response to TGFβ stimulation is dependent on Imp7[25, 26] (Supplementary Fig. 8a, b). First, we tested whether YAP could regulate Smad3 nuclear translocation upon TGFβ stimulation. YAP depletion with two independent siRNAs induced a significant increase in Smad3 nuclear accumulation upon TGFβ stimulation in comparison to control cells (Fig. 9a, b). TGFβ stimulation results in Smad3 serine phosphorylation, which precedes its nuclear translocation[105]. However, YAP did not regulate TGFβ-induced Smad3 phosphorylation, as YAP depleted cells had unchanged levels of phospho-Smad3 in basal conditions and after TGFβ stimulation, which induced phosphorylation as expected (Fig. 9c, d). YAP and TAZ have been shown to form complexes involving Smad3 in the nucleus, which could induce its nuclear retention[35, 39]. However, we did not observe any significant compensatory upregulation in the levels of TAZ upon YAP knockdown that could explain these effects (Fig. 9c, e). Thus, these observations suggest that other mechanisms independent of Smad3 phosphorylation and TAZ levels mediate the inhibitory effect of YAP on Smad3 nuclear accumulation. To test whether the increase in Smad3 nuclear accumulation was a consequence of increased binding between Smad3 and Imp7, we quantified the association of the two proteins by PLA upon YAP silencing. The Imp7–Smad3 association was specific, as silencing of either Imp7 or Smad3 effectively depleted the PLA signal (Supplementary Fig. 9a, b). The addition of TGFβ induced a robust increase in the PLA signal (Supplementary Fig. 9c, d), as expected. Upon YAP knockdown with two independent siRNAs and TGFβ stimulation, the Imp7–Smad3 complex significantly increased, as compared to cells silenced with control siRNA (Fig. 10a, b). These results suggest that YAP absence favors the association of Imp7 with Smad3 and its nuclear accumulation. To explore whether this phenotype was specific for YAP or other Imp7 cargo depletion has similar consequences on the binding of Imp7 to cargo, we performed the reverse experiment, i.e., we silenced Smad3 and monitored the association of Imp7 with YAP. As shown in Supplementary Fig. 9e, f, Smad3 silencing did not alter the association of Imp7-YAP. Consistent with this result, Smad3 silencing did not affect the nucleo-cytoplasmic balance of YAP (Supplementary Fig. 9g, h). Taken together, these results suggest that YAP reduces the association of Imp7 with Smad3 and its nuclear accumulation in response to TGFβ. Interestingly, this effect is unidirectional, as Smad3 does not induce any alteration of these patterns in YAP.

Our results suggest that the inhibitory effect of YAP on Smad3 nuclear translocation could be driven by the interaction between YAP and Imp7. Based on this rationale, YAP should not have any impact on the Imp7–Smad3 association is highly confluent cells, as YAP is inhibited by the Hippo pathway and does not interact with Imp7 (Fig. 7b, e). As shown in Fig. 10c, d, YAP silencing did not have a significant effect on Imp7–Smad3 association in highly confluent cells, as opposed to sparse cells (Fig. 10a, b).

As an independent approach to determine that the binding of YAP to Imp7 was competing with Smad3 binding to the same NTR, we overexpressed YAP wild type and mutants in low confluence cultures and analyzed their inhibitory effect on Imp7–Smad3 association. EYFP-YAP overexpression significantly reduced the Imp7–Smad3 complex, compared to control cells (Fig. 10e, f). In contrast, expression of EYFP-YAP S127E or EYFP-YAP ΔC, which are unable to directly and efficiently bind Imp7 (Figs. 4c, 7i, Supplementary Figs. 7a, b, d), did not have any effect on Imp7–Smad3 association (Fig. 10e, f), despite equal levels of expression respect to wild-type YAP (Fig. 10g),

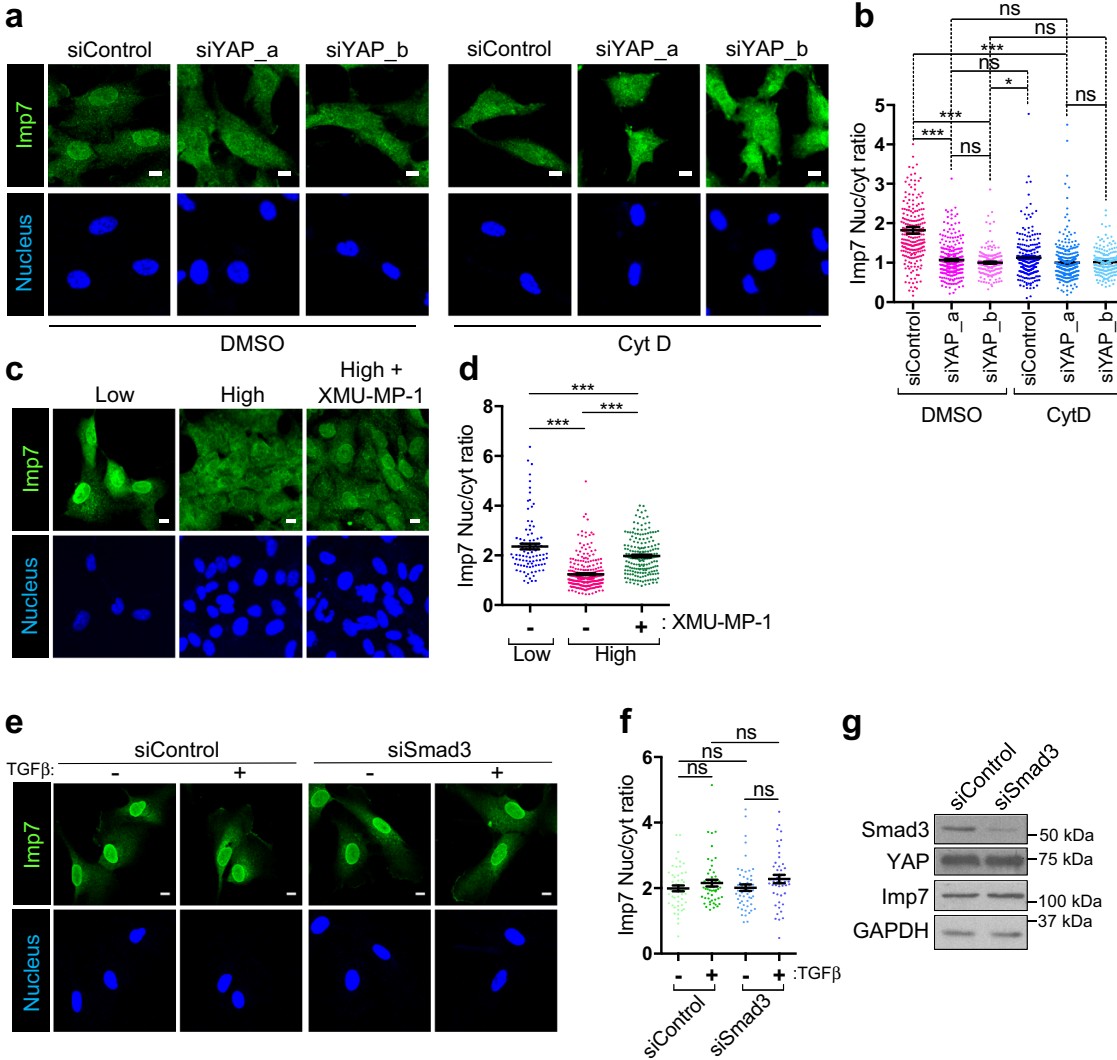

**Fig. 8 YAP determines Imp7 mechanoresponse. a**, **b** Immunofluorescence of endogenous Imp7 and nuclei in RPE-1 cells growing at the low confluence and silenced for YAP with two independent siRNAs. Vehicle (DMSO) or Cyt D (1 μM, 1 h) treated sets are shown. Quantification is shown in graph (**b**). From left to right in the graph, N = 240, 236, 270, 321, 167, and 190 cells per condition, from 3 independent experiments. P-value = 1.069e−14 (DMSO siControl vs DMSO siYAP_a), 1.210e-16 (DMSO siControl vs. DMSO siYAP_b), 5.361e-12 (DMSO siControl vs. Cyt D siYAP_a), 0.101 (DMSO siYAP_a vs. DMSO siYAP_b), 0.206 (DMSO siYAP_a vs. Cyt D siControl), 0.090 (DMSO siYAP_a vs. CytD siYAP_a), 0.011 (DMSO siYAP_b vs. Cyt D siControl), 0.831 (DMSO siYAP_b vs. Cyt D siYAP_b), and 0.804 (Cyt D siYAP_a vs. Cyt D siYAP_b). **c**, **d** Immunofluorescence of endogenous Imp7 and nuclei in RPE-1 cells growing at low or high confluence treated with 6 μM XMU-MP-1 for 16 h. Quantification is shown in graph (**d**). From left to right in the graph, N = 93, 273, and 194 cells per condition, from 3 independent experiments. P-value = 5.480e−14 (1st vs. 2nd bars), 0.0036 (1st vs. 3rd bars), and 1.422e−25 (2nd vs. 3rd bars). **e**, **f** Immunofluorescence of endogenous Imp7 in RPE-1 cells growing at a low confluence, silenced for Smad3 and treated with (+) or without (−) TGFβ (10 ng/ml for 2 h). Quantification is shown in graph (**f**). From left to right in the graph, N = 51, 50, 53, and 45 cells per condition, from 3 independent experiments. **g** Immunoblot showing specific depletion of Smad3. Several loading controls are shown. Representative of three biologically independent experiments. Raw data are available in the Source Data file. Statistical analysis with a two-tailed unpaired t-test. Data represent mean ± s.e.m. Scale bar 10 μm. ns non-significant. P-values below or equal to 0.05, 0.01, or 0.005 were considered statistically significant and were labeled with 1, 2, or 3 asterisks, respectively.

suggesting that the direct binding of YAP to Imp7 regulated the association of Imp7 with Smad3.

To further implicate the Hippo pathway in regulating Imp7-Smad3 association, we inhibited MST1/2 in highly confluent cells, which restores the association of YAP with Imp7 (Fig. 7e). Hippo pathway inhibition significantly reduced Imp7-Smad3 complex formation (Fig. 10h, i). Collectively, these results suggest that the interaction of YAP with Imp7, which only occurs when cell tension is high, restricts the association of Smad3 with Imp7, reducing its nuclear accumulation. Thus, a regulation between the Hippo and TGFβ pathways is achieved by a competition of one

effector over the other for their common NTR, which is in turn modulated by cell density (Fig. 10j).

This competition is not restricted to Smad3. Erk2 has been previously shown to be imported by Imp7[27, 98, 103] and, as expected, Imp7 depletion prevents nuclear accumulation of Erk2 (Supplementary Fig. 9i, j). Interestingly, YAP silencing increased the amount of Erk2 in the nucleus (Supplementary Fig. 9k, l), similar to Smad3 (Fig. 9a, b). Notably, the association between Erk2 and Imp7 increased upon YAP silencing (Supplementary Fig. 9m, n). Taken together, these results suggest that the YAP/Imp7 complex allows regulating other pathways by restricting the binding to Imp7.

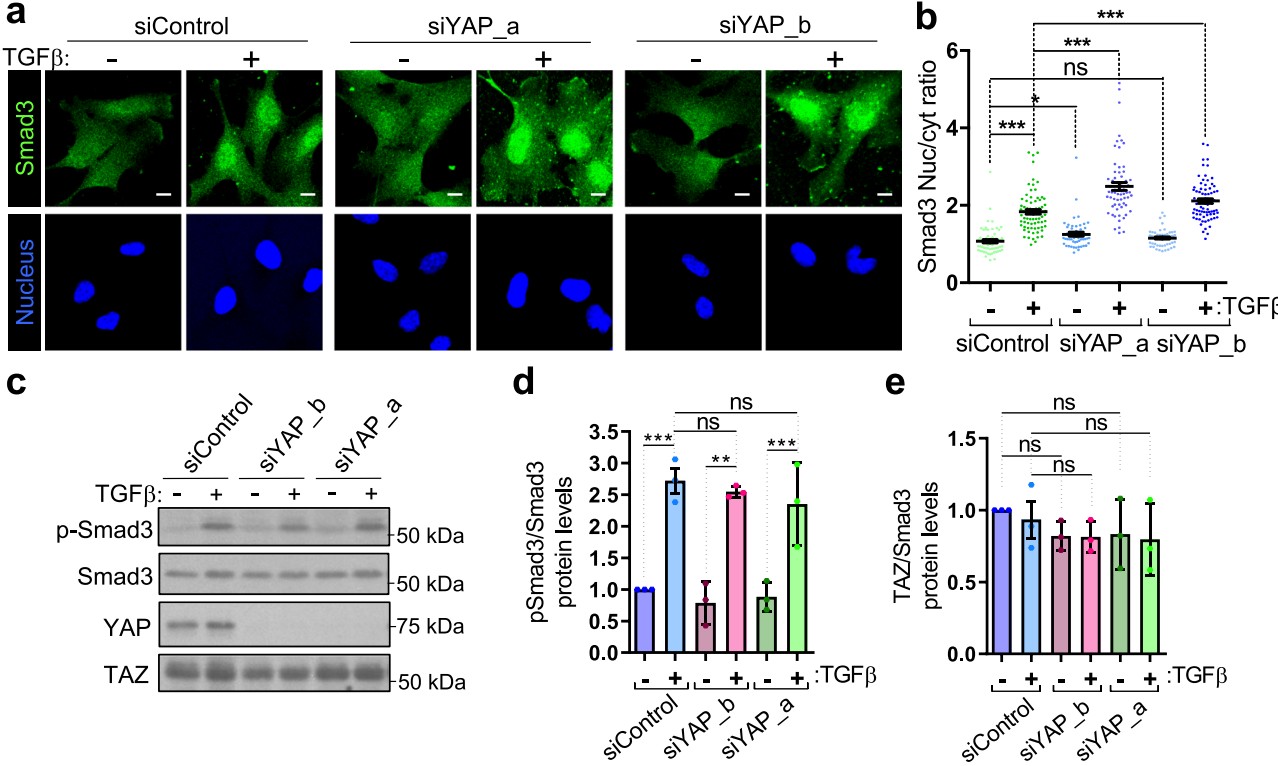

**Fig. 9 YAP restricts Smad3 nuclear accumulation independent of pSmad3 and TAZ. a**, **b** Immunofluorescence of endogenous Smad3 in RPE-1 cells at the low confluence and silenced for YAP with two independent siRNAs. Cells treated with (+) or without (−) TGFβ. Quantification is shown in graph (**b**). From left to right in the graph, N = 65, 77, 51, 64, 55, and 73 cells per condition, from 3 independent experiments. P-value = 2.356e−19 (1st vs. 2nd bars), 0.010 (1st vs. 3rd bars), 0.109 (1st vs. 5th bars), 1.147e−07 (2nd vs. 4th bars) and 0.00099 (2nd vs. 6th bars). **c**–**e** Immunoblot showing specific phosphorylation of Smad3, Smad3, YAP, and TAZ total levels in cells silenced for YAP with two independent siRNAs. Quantification of pS423/425 Smad3 (pSmad3)/Smad3 levels is shown in graph (**d**). N = 3 biologically independent experiments. Quantification of TAZ/Smad3 levels is shown in graph (**e**). N = 3 biologically independent experiments. Statistical analysis with a two-tailed unpaired *t*-test. Data represent mean ± s.e.m. In graph **d**, p-value = 0.0012 (1st vs. 2nd bars), 0.470 (2nd vs. 4th bars), 0.446 (2nd vs. 6th bars), 0.0083 (3rd vs. 4th bars) and 0.0047 (5th vs. 6th bars). In graph **e**, p-value = 0.095 (1st vs. 3rd bars), 0.356 (1st vs. 5th bars), 0.515 (2nd vs. 6th bars) and 0.464 (2nd vs. 4th bars). Scale bar 10 µm. ns non-significant. Raw data are available in the Source Data file. P-values below or equal to 0.05, 0.01, or 0.005 were considered statistically significant and were labeled with 1, 2, or 3 asterisks, respectively.

## Discussion

Nuclear translocation of transcriptional regulators is a rate-limiting step that must be overcome to regulate gene expression[105]. This is particularly challenging for large proteins that require the assistance of NTRs to translocate through the nuclear pore complex[20]. The human genome encodes for 20 Importin-β related NTRs and 7 Importin-α, of which around 19 are involved in the nuclear import of a variety of cargoes[20]. The reasons behind the abundance of different import factors are not entirely clear, but likely reflect different requirements of the transported cargoes. Using a quantitative proteomic approach, we identified Imp7 as an NTR highly responsive to cell density (Fig. 1a, b). Further analysis of this behavior showed that Imp7 accumulates predominantly in the nucleus in response to various mechanical inputs (Fig. 2) and cell tension-controlling pathways (Fig. 3). This behavior has not been reported for any other NTR, nor was it observed for those analyzed other than Imp7. The mechanoresponse capacity of Imp7 may suggest that Imp7 is designed to translocate nuclear proteins requiring regulation by mechanical cues; this is the case for YAP, and, in an indirect manner, also for Smad3 and Erk2, as their association with Imp7 is regulated by YAP (Fig. 10). Genetic studies in *Drosophila* have shown that Imp7 homolog Msk genetically interacts with genes encoding mechanoresponsive proteins[68, 106], such as integrins or the dystroglycan–dystrophin complex[103, 107–109], known to be

upstream of YAP[49, 58, 110, 111]. Although several Imp7 cargoes have ties to the Hippo pathway or have been linked to mechanobiology[25, 27, 98, 104, 112–114], other Imp7 cargoes have not been linked to mechanotransduction pathways[23, 24, 115]. Therefore, further investigation is needed to determine how general the involvement of Imp7 in regulating mechanotransduction is.

Conditions that preserve tension in the cell allow the formation of the Imp7–YAP complex (Fig. 7) and this is achieved by actin cytoskeleton- and cell confluence-dependent inhibition of MST kinases and/or upstream of them, which prevents YAP serine 127 phosphorylation and interaction with Imp7 (Fig. 10j). This scenario is fully compatible with the current understanding of the Hippo pathway functioning downstream of mechanical cues, whereby stress fibers inhibit LATS or MST kinases (or their *Drosophila* homologs Warts and Hippo) or upstream nodes[47–50, 54, 94]. The favorable Imp7–YAP interaction without an active Hippo pathway, i.e., lack of YAP phosphorylation, could explain why in vitro, where we used proteins purified from bacteria without post-translational modifications, the Imp7–YAP complex is formed and is fully competent for nuclear import in permeabilized cells and FG-barrier extravasation (Fig. 4). Therefore, the absence of tension-generating signals in vitro is not a limitation that prevents a functional nuclear translocation of the Imp7–YAP complex. The impact of cell tension pathways upstream of the Imp7–YAP complex formation is evident in cells

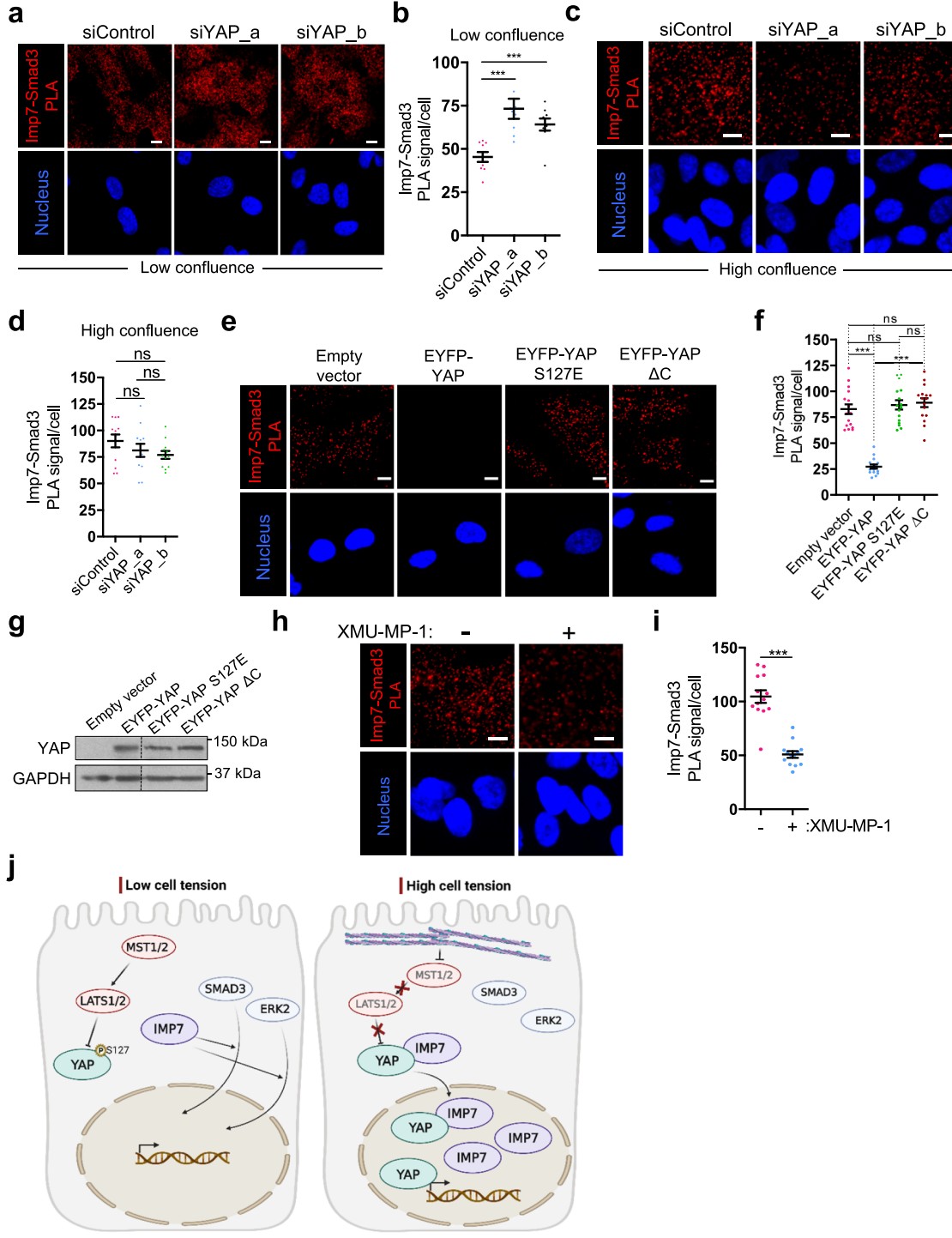

(Fig. 7), but we cannot rule out that mechanical signals, functioning downstream of the Imp7–YAP complex, may also play a role in facilitating nuclear translocation in cells. In vitro, the absence of these pathways could also be beneficial for the translocation, as it occurs with the lack of phosphorylation on YAP in vitro (Fig. 4). Therefore, our results in vitro, in cellulo, and in vivo are compatible with other pathways acting upstream or downstream of the YAP–Imp7 complex[90, 116–123].

In mammalian cells, the interplay between YAP/TAZ and Smads is quite complex[46]. YAP/TAZ can play positive or negative roles in R-Smad-mediated signaling[35, 38–40, 43–45]. In Drosophila, similar trends are observed, some of the Mad (R-Smad in

Drosophila) targets are co-regulated (e.g., ban gene) or inhibited (e.g., omb gene) by active Yki in vivo[124]. These studies suggest that, depending on the context, Hippo and TGFβ effectors may have a distinct interrelationship. The effect of YAP on Imp7–Smad3 binding is also highly dependent on the context: in highly confluent cells, YAP does not have any impact on Smad3 binding to Imp7, while in sparse cells it restricts the association (Figs. 9, 10 and Supplementary Fig. 9). Thus, cell density is key to regulate YAP-dependent Imp7–Smad3 association. Indeed, in various non-polarized cell types, TGFβ stimulation results in higher PAI-1 expression—a target of TGFβ—in highly confluent cells compared to low confluent cells, despite having the same

**Fig. 10 YAP restricts Smad3 and Imp7 association. a–d** In situ PLA detection of the association between endogenous Smad3 and Imp7 in RPE-1 cells growing at low (**a**, **b**) or high confluence (**c**, **d**) and silenced with control or two independent siRNAs for YAP. Quantification of the PLA signal is in panels (**b**) and (**d**). $N = 9$ fields (low confluence) and $N = 12$ fields (high confluence) from 3 independent experiments. In panel **b**: P-value = 0.0011 (1st vs. 2nd lanes) and 0.00079 (1st vs. 3rd lanes). In panel **d**: p-value = 0.311 (1st vs. 2nd lanes), 0.076 (1st vs. 3rd lanes), and 0.563 (2nd vs. 3rd lanes). **e**, **f** In situ PLA detection of the association between endogenous Smad3 and Imp7 in RPE-1 cells overexpressing the indicated proteins and growing at the low confluence. **f** Quantification of the PLA signal. $N = 16$ fields in each sample from 3 independent experiments. P-value = 8.563e−10 (1st vs. 2nd bars), 0.541 (1st vs. 3rd), 0.336 (1st vs. 4th), 1.437e−11 (2nd vs. 4th), and 0.726 (3rd vs. 4th). **g** Immunoblot showing the levels of overexpressed EYFP-YAP, EYFP-YAP S127E, and EYFP-YAP ΔC mutants in RPE-1 cells. **h**, **i** In situ PLA detection of the association between endogenous Smad3 and Imp7 in RPE-1 cells growing at the high confluence and treated with (+) or without (−) XMU-MP-1 for 16 h. **i** Quantification of the PLA signal. $N = 13$ fields for each sample from 3 independent experiments. Statistical analysis with a two-tailed unpaired $t$-test. Data represent mean ± s.e.m. Scale bar 10 μm. P-value = 2.014e−07. Raw data are available in the Source Data file. P-values below or equal to 0.05, 0.01, or 0.005 were considered statistically significant and were labeled with 1, 2, or 3 asterisks, respectively. **j** A graphical model to explain the results of this study. In conditions of low cell tension, such as high cell density or disruption of the actin cytoskeleton, YAP is no longer competent to bind Imp7 due to phosphorylation on YAP serine 127. This setting favors Smad3 and Erk2 association with Imp7, which results in more nuclear Smad3 and Erk2. On the contrary, under conditions of high cell tension, such as low cell density and intact contractile actin cytoskeleton, the Hippo pathway is inactive, resulting in the formation of the Imp7–YAP complex, which drives YAP and YAP-dependent Imp7 nuclear localization. Under these conditions, the association of Smad3 and Erk2 to Imp7 is limited by YAP. This mechanism permits the regulation of other signaling pathways by the Hippo pathway in a cell tension-dependent manner by the competition of YAP for Smad3 or Erk2 access to Imp7. Created using Biorender.com.

level of phospho-Smad3 in both conditions[45]; this is consistent with the presence of a stimulatory mechanism downstream of phospho-Smad3 in highly confluent cells, consistent with our findings.

YAP has been shown to form complexes in the nucleus involving Smad2/3, at least in malignant mesothelioma cells with deleted LATS[35]. Although it is not clear whether YAP directly binds Smad3 in the nucleus, co-immunoprecipitation experiments using overexpressed YAP and Smad3 showed that YAP association with Smad3 is through the WW domain (in the amino terminus). In contrast, the inhibition of YAP on Smad3–Imp7 complex is dependent on the last 5 amino acids of YAP (YAP ΔC, Figs. 10e, f and 4c), which suggests that YAP domains mediating Smad3 association and Smad3–Imp7 complex disruption map to different regions. This may also reflect that they are independent events[35]. Thus, YAP functionally interacts with Smad3 at least at two different levels: (i) by interfering with its association with Imp7 and (ii) by forming different complexes together with other transcriptional regulators in the nucleus[35, 36].

Smad3 can also be imported into the nucleus independently of Imp7[125, 126] but requires it upon TGFβ stimulation[25], which highlights that the nuclear translocation step per se is a highly regulated and versatile process that could be relevant in context-dependent gene expression. It is currently unclear whether these different nuclear entry modes have different functional outputs.

Similarly, YAP/Yki has been shown to interact with other importins in overexpression-based systems[85, 86]. In *Drosophila*, Importin-α1 was involved in regulating nuclear Yki accumulation; however, this study was based on overexpression approaches, did not show direct nor endogenous interaction between Yki and Importin-α1, nuclear import assays were not performed, and did not observe any effect when the mammalian Importin-α1 homolog was overexpressed[85]. Furthermore, when the effects of overexpressed Importin-α1 were tested in vivo, no effects on Yki wild type were observed[85]. Our screening on all mammalian importins showed that Importin-α1, Importin-α5, and Importin-α8 slightly prevented YAP nuclear accumulation (Fig. 5f, g). Interestingly, a study also based on overexpression systems detected the interaction between these same importins and YAP, but no additional approaches were used[86]. These results suggest that either Imp7-independent alternative entry routes exist for nuclear YAP or indirect effects on other YAP regulators, that do require those importins, are responsible for these observations. The fact that simultaneous depletion of all importins affecting YAP nuclear amount did not produce an accumulative

phenotype, favors the idea that the moderate effects of silencing Importin-α1, α5, and α8 on nuclear YAP are due to indirect causes. Indeed, indirect effects on nuclear YAP localization, independent of the nuclear import process, have been reported for classical NLS containing YAP regulators[121]. Nevertheless, further studies are needed to define the exact mechanism by which Importin-α1, α5, and α8 regulate nuclear YAP amount. YAP lacks a classical NLS, while a region resembling a recently proposed NLS for Imp7[96] is essential to bind Imp7 and to accumulate YAP in the nucleus (Supplementary Fig. 7c–g), suggesting that it may serve, de facto, as an NLS. However, more studies are needed to define this sequence as a general NLS for Imp7.

In the case of TAZ, a recent study showed that it does not require active nuclear transport by an NTR[74]. TAZ runs about 20 kDa below YAP (Fig. 9c), which might explain the differences in the nuclear translocation requirements.

The PDZ-binding motif in YAP/TAZ, which mediates binding to some PDZ containing proteins[116, 127], and an analogous region in Yki, are also essential to bind Imp7/Msk. However, we did not find a PDZ domain in Imp7/Msk using standard domain search tools, suggesting that the last amino acids of YAP/Yki may function in a unique manner and are slightly different from other PDZ-binding motifs. Another requirement for the interaction to occur is the presence of unphosphorylated serine 127, which together with the PDZ-binding motif and the NLS7 is essential to bind Imp7 and efficiently accumulate in the nucleus (Supplementary Fig. 7).

The identification of the mechanism by which YAP translocates into the nucleus and the regions in YAP responsible for the interaction with Imp7 open the possibility of intervening in this process by preventing YAP binding to Imp7. The fact that Imp7/Msk silencing is sufficient to block YAP/Yki-induced aberrant tissue overgrowth in vivo (Fig. 6) supports this rationale. This could have therapeutic applicability in several pathologies, including fibrosis, atherosclerosis, and different types of cancer, where the aberrant increase in nuclear YAP localization is associated with the disease[28, 56, 128, 129]. It will be interesting to define how the three important regions defining the interaction YAP/Imp7 are organized in the 3D structure of YAP. This will allow us to fully understand how the contact between these two molecules occurs, which will contribute to designing strategies to prevent the interaction for therapeutic purposes.

The data presented here suggest a model where tension variations can modify the extent of YAP nuclear translocation by

modulating its binding to Imp7. This simultaneously controls other pathways by limiting the access to Imp7 (Fig. 10j). The effect of YAP over other cargoes is not reciprocal, at least in the case of Smad3, suggesting that YAP may play a dominant role over the access to Imp7 by other cargoes, which could position YAP as a potential regulator of the mechanical sensitivity of multiple pathways. This represents a distinct mechanism of nuclear import regulation and mechanotransduction pathways crosstalk.

## Methods

**Cell culture, reagents, and plasmids.** RPE-1 cells (ATCC CRL-4000) were grown in DMEM/F-12 (Lonza) and MOVAS (ATCC CRL-2797) and HEK293 (ATCC CRL-1573) cells were grown in DMEM. Growth media was supplemented with 10% fetal bovine serum (FBS; Thermo Fisher Scientific), 2 mM glutamine, and 100 μg/ml penicillin and streptomycin (Thermo Fisher Scientific). MSCs from human donors were purchased from SIGMA and cultured in MSC expansion medium (Millipore SCM015) supplemented with 10% FBS, 2 mM glutamine, and 8 ng/ml hbFGF (F0291, SIGMA). Cell lines were maintained in a humidified atmosphere at 37 °C and 5% $CO_2$. S2 cells (ATCC, CRL-1963) were grown on Schneider´s *Drosophila* medium (Gibco). Media was supplemented with 10% heat-inactivated fetal bovine serum (Thermo Fisher Scientific) and 100 μg/ml penicillin and streptomycin (Thermo Fisher Scientific). Cells were maintained at 25 °C. Plasmids were transfected into HEK293 cells using Fugene6 (Roche), into RPE-1 cells using either Lipofectamine 2000 or Lipofectamine 3000 (Invitrogen), and into S2 cells using Lipofectamine 3000. siRNAs were transfected at 20 nM with RNAiMAX (Invitrogen). For MSCs RNAi experiments, a double transfection protocol was performed, with a second hit two days after the first one. Rabbit-made antibodies: GST (Invitrogen #717500; western blot (WB) 1:2000), pS127 YAP (Cell Signaling #4911S; WB 1:1000), YAP (Cell Signaling #14074; IF 1:100), Imp7 (Proteintech #28289-1-AP; IF 1:100), Smad3 (Cell Signaling Technology #9523; WB 1:1000; IF 1:100), pSmad3 S423/425 (Cell Signaling Technology #9520S), Erk2 (Santa Cruz, sc-154; IF 1:100), Yki (kindly provided by Dr Keneth Irvine[90]; IF 1:100), YAP (Proteintech 13584-1-AP; IF 1:200). Mouse-made antibodies: Imp7 E-2 (Santa Cruz Biotechnology, sc-365231; WB 1:1000; IF 1:200), YAP 63.7 (Santa Cruz, which recognizes YAP and TAZ[18], sc-101199; WB 1:1000; IF 1:100), YAP H-9 (Santa Cruz sc271134; IF 1:50), GAPDH (Millipore, MAB374; WB 1:10000), HA (Roche, #11583816001; WB 1:1000), GFP (Roche #11814460; WB 1:1000; IF 1:500), ESE-1 (Santa Cruz E-8 sc-376055; IP 1:200). Guinea pig-made antibody: Msk (kindly provided by Prof. E. Geisbrecht[130]; IF 1:100). Uncropped and unprocessed WBs are shown in the source data file. siRNA oligonucleotides were purchased from Dharmacon and Ambion. As control non-targeting siRNA, control #1 (Dharmacon) was used. From Ambion: siRNA duplexes targeting IPO7 (encoding Imp7): (#s20640 (siImp7_a) (GGAAUCUGCUUACAGGUCAtt) and IPO7 #s20638 (siImp7_b) (GACUGACAAGAGAGGGUUAAtt)); RAN (#s11769, GGAUAAUUAAGGACAGGAAAtt); Smad3 (#s535081, GGUGCUCCAU-CUCCUACUAtt); YAP (#s534572 (siYAP_b), AGAUACUUCUUAAAUCACAtt); KPNB1 (encoding Impβ1) (#s7919, CAGUGUAGUUGUUCGAGAUtt); KPNA1 (encoding Impα1) (#s223980, GCUUGGGUACUGACAAAUAtt); KPNA2 (encoding Impα5) (#s7922, GAGACUUGGUUAUUAAGUAtt), IPO11 (encoding Imp11) (#s27653); KPNA5 (encoding Impα6) (#s7931); IPO5 (encoding Imp5) (#s7936); TNPO1 (encoding Transportin-1) (#s7933); IPO9 (encoding Imp9) (#s31299); KPNA6 (encoding Impα7) (#s24242); KPNA4 (encoding Impα3) (#s7928); XPO4 (encoding Exportin-4) (#s34638); IPO13 (encoding Importin-13) (#s18608); IPO4 (encoding Importin-4) (#s36154); TNPO3 (encoding Transportin-3) (#s532942); TNPO2 (encoding Transportin-2) (#s26880); IPO8 (encoding Importin-8) (#s20653). SiGENOME SMARTpool siRNA against YAP (siYAP_a) (L-012200-00-0005) was purchased from Dharmacon. Preselect Mission esiRNA against KPNA7 (encoding Impα8) was purchased from Sigma. pEF-HA Imp7 (*X. laevis*) plasmid was a kind gift provided by Dr. Ralph H. Kehlenbach[131]. p2×Flag CMV-YAP2 (Addgene #19045). pCMV-Flag YAP2 5SA (Addgene #27371). 8× GTIIC-luciferase (Addgene #34615). pGEX-KG-GST-YAP (Addgene #33052). GST-Imp7 expressing plasmid (pGEX6P-Imp7) was kindly provided by Ariberto Fassati. pLVX-CMV-CherryFP-P2A-MetLuc was described previously[73]. The GFP-DN KASH plasmid was a gift from Catherine Shanahan[70]. UAS-Msk-YFP was a gift from Dr. Erika Geisbrecht[92]. pCMV-Flag YAP2 5SA (Addgene #27371). pUAST-Yki:Flag(x3) (Addgene #85622). pUAST-Yki S168A:Flag(3×) (was a gift from Dr. Keneth Irvine[90]. pUAST-Yki ΔC:Flag(3×) was created using HiFi Assembly (NEB) and lacks the last 7 residues of Yki (sequence LEWYKIN). pLL3.7-EF-EYFP-YAP1_delta5C-PolyA (Addgene #112289). pLL3.7-EF-EYFP-YAP1_WT-PolyA (Addgene #112284). pLL3.7-EF-EYFP-YAP1_ΔRK[2] contains these mutations: R88, K89, R203, and K204 to A. pLL3.7-EF-EYFP-YAP1_ΔTPT contains these mutations: T143, P144, T145, T154, P155, T156, T412, P413, and D414 to A. pLL3.7-EF-EYFP-YAP1_S127E and EF-EYFP-YAP1_S128E and the previous mutations were created by directed mutagenesis (QuikChange Lightning Site-Directed Mutagenesis Kit, Agilent). pLL3.7-EF-EYFP-YAP1_ΔNLS7 was made using HiFi Assembly (NEB) to change the putative NLS7 sequence "EKERLRL" (positions 314–320) to NAAIRSA. EYFP-YAP1 from pLL3.7-EF-EYFP-YAP1 was subcloned into pEGFP-N1 (NheI/EcoRI), where mutagenesis was performed and then the mutated EYFP-YAP1 was cloned back into pLL3.7-EF. YAP1 in pLL3.7-EF plasmid corresponds to NM_001130145.3.

**Polyacrylamide matrices.** Hydrogels were prepared with different poly-acrylamide:bisacrylamide concentrations to reach a rigidity of 2.3 and 55 kPa as indicated[132]. The hydrogels were coated with fibronectin.

**Cell treatments.** For mechanical stretching, 24 h after plating on pronectin-coated 6-well silicon plates, cells were subjected to bi-axial constant stretching (20% amplitude) for 1 h on a programmable Flexcell FX-5000TT Tension System (FlexCell) under standard culture conditions. 0% of stretching was used as control.

Drugs were added to cells 24 h after platting. The ROCK inhibitor Y27632 (Y0503, added for 1 h at 25 μM), Blebbistatin (B0560, added for 1 h at 10 μM), Cytochalasin D (C8273, added for 1 h at 1 μM) and XMU-MP-1 (SML2233, added for 16 h at 6 μM) were from Sigma-Aldrich.

For spreading experiments in S2, cells were seeded on top of concanavalin A-coated surfaces. The coating was performed using ConA 0.5 mg/ml overnight at 37 °C. Cytochalasin D was added at 1 μM for 1 h to the media.

**Immunoprecipitation.** For immunoprecipitation of endogenous YAP and Imp7 or over-expressed YFP-Msk, cells were lysed in IP buffer (50 mM Tris-HCl pH 8.0, 100 mM NaCl, 1% Triton X-100 (Sigma-Aldrich), 10% glycerol, 1 mM $MgCl_2$, 2 mM PMSF, and protease inhibitor cocktail (Roche)). Cell lysates were centrifuged for 10 min at 4 °C. Supernatants were mixed with the primary antibody for 2 h and protein G-agarose beads for 2 more hours. Beads were washed with washing buffer (50 mM Tris-HCl pH 7.5, 150 mM NaCl, 1 mM EDTA, 0.25% gelatin, 0.1% NP-40) and processed for SDS-PAGE. For western blotting, protein extracts were lysed with sample buffer and analyzed by western blotting on nitrocellulose membranes (Amersham Pharmacia Biotech, UK) using secondary HRP-conjugated antibodies. Proteins were detected by enhanced chemiluminescence (Amersham Life Sciences).

**Immunofluorescence, microscopy, and image analysis.** Images were acquired on a Zeiss LSM 700 or a Leica TCS SP5 (confocal). ImageJ/Fiji was used to adjust brightness and contrast values. For immunostaining, cells were fixed with 4% paraformaldehyde at 37 °C for 10 min, washed 3 times with PBS, permeabilized with 0.2% Triton X-100 for 5 min, blocked with BSA 2% or FBS, incubated with primary antibodies 1 h at room temperature (except for YAP, overnight 4 °C), and incubated with secondary antibodies (1 h, room temperature). Nuclei were counterstained with Hoechst 33342. Glass slides with pre-printed micropatterns were purchased from Cytoo (Grenoble, France). Fibronectin coating was performed as specified by the supplier. Cells were plated, and 24 h later, fixed and stained. Nuclear to cytosol ratios were calculated selecting an area in the nucleus and adjacent to it an equal area in the cytosol was selected. The partition coefficient was calculated as the integrated intensity in the nuclear area divided by the cytosolic area. For PLA, cells were fixed with PFA 4% for 10 min, followed by 5 min per-meabilization with 0.2% Triton X-100. After the PBS wash, the manufacturer´s protocol was followed (Duolink In Situ Orange Kit Mouse/Rabbit, Sigma Aldrich). For quantification, RGB images were adjusted to a Maximum Entropy Threshold (Fiji) and the obtained images were submitted to the "Analyze Particles" plugin in Fiji. Images in PLA experiments show red puncta within cells, which represent the PLA assay signal. Results were analyzed in Microsoft Excel.

**Luciferase assay.** Luciferase assays to monitor TEAD transcriptional activity with the 8× GTIIC-luciferase reporter were as described[18]. Cells were transiently co-transfected with 8× GTIIC-luciferase (product: firefly luciferase) and pLVX-CMV-CherryFP-P2AMetLuc (product: Metridia luciferase [MelLuc], which is secreted to the medium). Luciferase activity was monitored with the DualLuciferase Reporter Assay System (Promega; Madison, Wisconsin, United States) in an ORION II microplate luminometer (Titertek Berthold; Bad Wildbad, Germany). Firefly luciferase was quantified in cell lysates by adding Luciferase Assay Reagent II (LARII), and MetLuc was quantified in culture medium by adding Stop & Glo reagent. Firefly luciferase activity was normalized to MetLuc activity to control for variability in transfection efficiency across samples.

**Real-time quantitative PCR.** RNA was extracted from cell samples with the RNAeasy micro kit (QIAGEN; Hilden, Germany). For each sample, 1 μg RNA was reverse transcribed using the Omniscript RT kit (QIAGEN) and random primers (Promega). qRT-PCR was performed with SYBR green (Applied Biosystems; Foster City, California, USA). Appropriate negative and positive controls were used. Results were normalized to endogenous GAPDH, HPRT1 and ACTIN expression using Bio-Rad CFX Manager. Forward and reverse primers are described for each gene. GAPDH ATCACCATCTTCCAGGAGCG, CCTGCAAATGAGCCCCAG; HPRT1, CCTGGCGTCGTGATTAGTGAT, AGACGTTCAGTCCTGTCCATAA; ACTB, CACCTTCCAGCAGATGTGGA, AGCATTTG CGGTGGACGATGG; AN KRD1, AGTAGAGGAACTGGTCACTGG, TGTTTCTCGCTTTTCCACTGTT; CTGF, ACCGACTGGAAGACACGTTTG, CCAGGTCAGCTTCGCAAGG; YAP, AGGTTGGGGAGATGGCAAAGA, ACCTGAAGCCGAGTTCATCA.

**Protein purification**. For GST-Imp7 and GST purification, overnight growth cultured bacteria were diluted 1/10 with LB to a final volume of 2 L, followed by 30°C growth at 185 rpm to an $OD_{600}$ of 0.9–1.0. Expression was induced with 1% glycerol, 2.5% ethanol, and 0.25 mM IPTG for 5 h at 15°C. Afterward, cells were harvested and stored at −80 °C. After thawing, cells were resuspended in BA buffer (250 mM NaCl, 10% (v/v) glycerol, 50 mM Tris-HCl pH 7.4, 10 mM DTT, 2 mM EDTA, 0.5 mM PMSF). Cell lysates were sonicated and ultracentrifuged twice at 12,000 rpm for 20 min and an additional 10 min. Supernatants were collected and mixed with agarose/sepharose 4B beads (GE Healthcare). After 1 h rocking at 4 °C, beads were extensively washed with BA buffer. GST-Imp7 was digested O/N in incubation at 4 °C with pre-scission protease (GE) to obtain pure Imp7. Imp7 was dialyzed against 50 mM Tris-HCl pH 8.0, 100 mM NaCl, 1% Triton X-100 (Sigma-Aldrich), 10% glycerol, 1 mM $MgCl_2$ using 200 MW membrane cassettes (Thermo Scientific).

For GFP8Q-YAP (GFP8Q, abbreviated from efGFP8Q, is an engineered version of GFP, showing neutral interactions with the FG hydrogels[81] and, thus, suitable for tagging a target protein for nuclear transport assays), transformed *NEB Iq Expression* bacteria were grown overnight at 37 °C. The overnight culture was diluted with TB medium and grown at 37 °C until $OD_{600}$ achieved 0.8–1.0. Cultures were diluted 1:1 with TB medium and grown at 18 °C until $OD_{600}$ achieved 1.8–2.2. Expression was induced with 100 μM IPTG at 18 °C overnight. After centrifugation, cell pellets were resuspended in 25 mM Tris-HCl pH 8.0, 500 mM NaCl and frozen at -80 ˚C. After thawing, DTT was added to a final concentration of 2 mM. Lysates were then sonicated and ultracentrifuged at 38,000 rpm for 1 h at 4 °C. Supernatants were collected, 10 mM imidazole was added and they were mixed with Ni-beads for 1 h at 4 °C. Afterwards, the mix was washed in a cocktail with: (I) 25 mM Tris-HCl pH 8.0, 500 mM NaCl, 2 mM DTT, 30 mM imidazole; (II) 2 M NaCl, 0.5 mM ATP; and (III) 25 mM Tris-HCl pH 8.0, 250 mM NaCl, 2 mM DTT. After washing, 25 mM Tris-HCl pH 8.0, 250 mM NaCl, 2 mM DTT, brSUMO protease[133, 134] was added and incubated O/N. Elution was carried out with the same buffer without the protease. Fractions with GFP were collected, concentrated in Amicon Ultra Centrifugal Filters (Merck Millipore) and gel filtrated in Superdex 200 column (GE Healthcare). Fraction aliquots were analyzed by SDS-PAGE and fractions with the protein of interest were selected. *Saccharomyces cerevisiae* (sc) Nup116 FG-domain was purified as before[80].

**GST pull-down assays**. Sepharose beads containing GST or GST-Imp7 were washed in IP buffer, mixed with YAP-GFP8Q, and incubated at 4 °C for 1.5 h. Samples were washed with washing buffer (50 mM Tris-HCl pH 7.5, 150 mM NaCl, 1 mM EDTA, 0.25% gelatin, 0.1% NP-40), loaded in SDS-PAGE, and stained with Coomassie.

**Cell permeabilized assay for nuclear import**. Permeabilized HeLa cells were used as followed: briefly, cells were grown on 10-well plates to ~50% confluence and permeabilized with digitonin (Sigma-Aldrich) for 3 min in a buffer containing 20 mM Hepes pH 7.5, 3.5 mM MgAc, 110 mM KAc, 1 mM EGTA and 250 mM sucrose at room temperature. After three washes in permeabilization buffer without digitonin, each well was incubated with 75 μl of import mixture containing 0.2 μM GFP8Q-YAP, 0.4 μM Imp7, Ran mix (RanGDP, RanGAP, RanBP1, and NTF2), 0.2 μM Atto647N-labeled MBP (G260C mutant), and energetic-mix (GTP, ATP, creatine phosphate and creatine kinase) and acquired in a Leica LSM 780 system.

**FG-particles assay**. Cargo and NTR were mixed and left at room temperature for 30 min. Afterward, a buffer containing 50 mM Tris-HCl pH 7.5, 150 mM NaCl was added to a final volume of 25 μl. The concentrations of each component in the mix were: 1 μM cargo and 2 μM NTR. In the plate well (μ-slide 18 well-Flat collagen IV, IBIDI), the mix was united to scNup116 FG-particles, left 30 min at room temperature in a dark and wet recipient, and analyzed by confocal microscope. FG particle preparation was as previously described[80, 81].

**Fly stocks and genetics**. All fly stocks were maintained at 25 °C (unless otherwise specified) on 12/12 h light/dark cycles at constant humidity in standard medium. The stocks used from Bloomington Stock Center were: UAS-LacZ (BL-8529) and UAS-msk RNAi (BL-23944). Other stocks were ap-Gal4,UAS-GFPs;tub-Gal80ts, and ap-Gal4,UAS-GFPs, UAS-yki tub-Gal80ts (gift from H. Herranz[135]). GFP-Yki (BL28815) flies express GFP-tagged Yki form under UAS control. Apterous-Gal4 (BL3041) flies express Gal4 under the control of *apterous* promoter were used to induce the expression of UAS constructs in the dorsal compartment of wing imaginal discs. UAS-GFP (BL1522) flies express GFP under the control of UAS. UAS-YFP-Msk flies were kindly provided by Erika Geisbrecht[92]. Msk-HA flies were purchased from Flyorf[136]. Ten female *apterous*-Gal4 flies were crossed by the corresponding UAS males for each experiment.

Gal4 is a transcription factor from *Saccharomyces cerevisiae* that recognizes the UAS (upstream activating sequences) promoting the expression of the genes downstream[137]. Ap-Gal4 line drives the expression to the dorsal compartment of the imaginal wing disc of *Drosophila* larvae from the second instar[138]. Gal80TS is a repressor of the Gal4 activity at 18 °C, though at 29 °C is inactivated[139]. The tub-Gal80ts construct was used in all the crosses to avoid the expression of the transgenes until the second instar larvae. To inactivate the Gal80ts protein and activate the Gal4/UAS system to allow the expression of our genes of interest, the second instar larvae were maintained at 29 °C for 48 h.

**Immunostaining and image acquisition in *Drosophila* experiments**. *Drosophila* larval wing discs were dissected and fixed with 4% formaldehyde in phosphate-buffered saline for 20 min. Samples were washed 3 × 15 min with PBS + 0.4% triton, blocked for 1 h with BSA 5%, incubated overnight at 4 °C with primary antibodies, washed 3 × 15 min, incubated with secondary antibodies for 2 h at room temperature, washed 3 × 15 min and mounted in Vectashield mounting medium with DAPI. The primary antibodies were anti-GFP rabbit (1:500, DSHB) and anti-Msk guinea pig (1:100). The secondary antibody used was Alexa 488 and Alexa 564 (1/200, Life Technologies). Images were taken by Leica SP5 confocal microscopy and the surface of the dorsal compartment (in green) and the total surface of the wing disc were analyzed using Image J. The ratio between compartments was analyzed by measuring the surface of the dorsal compartment (in green) and the total surface of the wing disc using Image J. The intensity of the anti-Msk staining was analyzed by measuring the mean pixel intensity of similar zones of both compartments using the sum of the slices.

**Quantitative proteomic analysis**

*Protein digestion and isobaric labeling.* For the quantitative differential proteomic analysis using isobaric tags (TMT 10-plex), about 200 μg of total protein from nuclear or cytosolic fractions were digested using the FASP protocol, with minor modifications. Samples were denatured by boiling for 5 min in the presence of 1% SDS and 50 mM DTT, diluted in 7 M urea in 0.1 M Tris-HCl (pH 8.5: UA buffer) and loaded onto 10 kDa centrifugal filter devices (NanoSep 10k Omega, Pall Life Sciences). Samples were alkylated for 45 min at 37 °C in the dark in 50 mM IAA (iodoacetamide) in UA buffer. The excess alkylating reagent was eliminated by washing two times with UA buffer and two more times with 100 mM ammonium bicarbonate. The proteins were then digested overnight at 37 °C with modified trypsin (30:1 protein:trypsin (w/w) in 100 mM ammonium bicarbonate: Promega), and the resulting peptides were twice eluted by centrifugation with 100 mM ammonium bicarbonate and 0.5 M sodium chloride. Trifluoroacetic acid (TFA) was added to a final concentration of 1%, and the peptides were desalted onto C18 Oasis-HLB cartridges and dried for further analysis.

For stable isobaric labeling, the tryptic peptides obtained were dissolved in 100 mM triethylammonium bicarbonate (TEAB) buffer and the peptide concentration was determined by measuring the amide bonds with the Direct Detect system (Millipore). Equal amounts of each peptide sample were isotopically-labeled, as indicated in Supplementary Table 2, by using the 10-plex TMT Reagents (Thermo Fisher) following the manufacturer's instructions. Basically, the peptides were labeled with the TMT reagents previously reconstituted in 42 μL of acetonitrile (ACN), and after incubation at room temperature for 2 h, the reaction was stopped by adding 0.5% TFA for 30 min. Internal standard consisted of a mixture of all samples in equal amounts. Finally, samples were concentrated in a Speed Vac, desalted onto C18 Oasis-HLB cartridges, and dried for further analysis.

*Protein identification and quantitation.* Labeled peptides were analyzed by LC–MS/MS using a C-18 reversed-phase nano-column (75 μm I.D. × 50 cm, 2 μm particle size, Acclaim PepMap RSLC, 100 C18: Thermo Fisher Scientific) and a continuous ACN gradient consisting of 0–30% B for 360 min, 50–90% B in 3 min (A = 0.1% formic acid; B = 90% ACN, 0.1% formic acid)- FA). A flow rate of 200 nL/min was used to elute peptides from the nano-column to an emitter nanospray needle for real-time ionization and peptide fragmentation on an HF Orbitrap mass spectrometer (Thermo Fisher). An enhanced FT-resolution spectrum (resolution = 60,000) and the MS/MS spectra from the Nth most intense parent ions were analyzed in the chromatography run. Dynamic exclusion was set at 40 s. For peptide identification, all the spectra were analyzed with Proteome Discoverer (version 2.1.0.81, Thermo Fisher Scientific) using SEQUEST-HT (Thermo Fisher Scientific).

The Uniprot database that contains all the sequences from mouse and human contaminants was searched, selecting the parameters: trypsin digestion with 2 maximum missed cleavage sites; precursor and fragment mass tolerances of 2 and 0.02 Da, respectively; TMT modifications at N-terminal and Lys residues as fixed modifications; and methionine oxidation, carbamidomethyl cysteine, and MMTS modified-cysteine as dynamic modifications (discovery phase). Peptide identification was performed using the probability ratio method[140], and the false discovery rate (FDR) was calculated using inverted databases and the refined method[141] with an additional filter for precursor mass tolerance of 15 ppm. The peptides identified had an FDR ≤ 1% and only these peptides were used to quantify the relative abundance of each protein from reporter ion intensities. For statistical analysis of the quantitative data, the previously described WSPP statistical model was used[142]. In this model, the protein log2-ratios are expressed as standardized variables, i.e., in units of standard deviation according to their estimated variances (Zq values).

**Statistics**. Statistical comparisons were carried out using a two-tailed unpaired Student's *t*-test. At least three independent experiments are analyzed for each condition. *P*-values below or equal to 0.05, 0.01, or 0.005 were considered

statistically significant and were labeled with 1, 2, or 3 asterisks respectively. In all figures, measurements are reported as mean ± standard error of the mean (s.e.m.).

**Reporting summary**. Further information on research design is available in the Nature Research Reporting Summary linked to this article.

## Data availability

The proteomic datasets used during the current study are available from a public repository. All the other data are available within the article and its Supplementary Information. The mass spectrometry proteomics data have been deposited to the ProteomeXchange Consortium via the PRIDE partner repository with the dataset identified PXD025910. Source data are provided with this paper.

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

## Acknowledgements

We thank Miguel Sánchez for text editing. We thank Erika R. Geisbrecht, Kenneth Irvine, and Ariberto Fassati for kindly providing reagents. This study was supported by grants from the Spanish Ministry of Science and Innovation (MICIIN)/Agencia Estatal de Investigación (AEI)/European Regional Development Fund (ARDF/FEDER) "A way to make Europe" (PID2020-118658RB-I00, SAF2017-83130-R, IGP-SO grant MINSEV1512-07-2016, CSD2009-0016 and BFU2016-81912-REDC), Comunidad Autónoma de Madrid (Tec4Bio-CM, S2018/NMT¬4443), Fundació La Marató de TV3 (201936-30-31), "La Caixa" Foundation (HR20-00075) and AECC (PROYE20089DELP) all to M.A.d.P. This project has received funding from the European Union's Horizon 2020 research and innovation program under the Marie Sklodowska-Curie grant agreement No. 641639. M.G.G. and L.S. are sponsored by FPU fellowships (FPU15/03776 and FPU18/05394, respectively). The CNIC is supported by the Instituto de Salud Carlos III (ISCIII), the Ministerio de Ciencia e Innovación (MICIIN) and the Pro CNIC Foundation, and is a Severo Ochoa Center of Excellence CEX2020-001041-S.

## Author contributions

A.E. and M.A.d.P. conceived/supervised the project, designed experiments, analyzed results, and wrote the paper together with M.G.G. S.C.T. and D.G. designed and analyzed experiments, edited the text, and contributed to the writing. M.G.G., E.C., and J.V. performed proteomic experiments that led to the identification of Imp7. P.J. and S.C.T. designed, performed, and analyzed fly experiments. M.G.G., A.E., S.S.P, T.H., L.F., and S.C.N. designed and performed experiments, and analyzed data together with L.S. and E.C.

## Competing interests

The authors declare no competing interests.
