## [Peer Review File · Nature Communications]

Mechanical control of nuclear import by Importin-7 is regulated by its dominant cargo YAPEditorial Note: Parts of this Peer Review File have been redacted as indicated to remove third-party material where no permission to publish could be obtained.

REVIEWER COMMENTS

Reviewer #1 (Remarks to the Author):

Garcia-Garcia et al

The manuscript investigates the role of the Importin-7 protein in nuclear import of YAP, and the ability of both proteins to be mechanically regulated in cell culture. The mechano-regulation of YAP has been the subject of intense investigation, and the current manuscript does not add significant new insight into the regulation of YAP. Rather, the novel results centre on Importin-7, which has long been known to participate in the nuclear import of multiple proteins, but has not been implicated in shuttling YAP through the nuclear pore. Thus, the identification of YAP as a cargo of Importin-7 is the main new finding. siRNA silencing of Imp7/IPO7 inhibits nuclear accumulation of YAP/TAZ, although only partially (see Fig S6f!). Similarly, siRNA silencing of YAP/TAZ, or inhibition of MST1/2 kinases, modulates the subcellular localisation of Imp7/IPO7, which suggests that the mechano-regulation of Imp7 may just be due to piggybacking on YAP/TAZ. Some further claims about competition between YAP/TAZ and other cargo of Imp7 are made, although the magnitude of the effect sizes are generally quite small in some of these experiments (e.g.: Fig S9I). The authors conclude that nuclear import 'is not a mere passive shuttling system, but an additional regulatory layer in which a cargo (YAP) directs the nuclear transport receptor Imp7 and competes out other cargoes'. It is claimed that this is a 'novel paradigm'.

Overall, my feeling is that the authors are trying too hard to explain everything about mechanotransduction with their favourite molecule, Imp7. The manuscript would be greatly improved by (1) changing the title to simplify it (Importin-7 promotes YAP nuclear import); (2) changing the abstract to make clear that the role of Imp7 is actually consistent with a passive shuttling system in which Imp7 may preferentially form a complex with YAP/TAZ over other cargoes; (3) changing the results/discussion to moderate some of their strong interpretations.

Major comments

1. Upon silencing of Imp7/IPO7, YAP is not excluded from the nucleus, being distributed at 1-1 nuclear/cytoplasmic ratio in the absence of Imp7, which raises the question of whether YAP/TAZ can also be a cargo for other Importins, just as Imp7 shuttles many different cargo. If so, it is difficult to see how the authors can conclude that Imp7 is a component of the Hippo pathway, or how Imp7 can be 'the explanation' for how YAP/TAZ is regulated by mechanical cues.
2. In *Drosophila*, silencing of Imp7 (*msk*) reduces imaginal disc size upon overexpression of UAS.Yki. This needs much further investigation. For example, what are the adult phenotypes of UAS.*msk*-RNAi in the eye and wing? Is UAS.*msk*-RNAi able to prevent Yki nuclear localisation in *Drosophila*? And is the subcellular localisation of Imp7/Msk also regulated in the same way as *Drosophila* Yki? These are essential questions to answer.
3. The Imp7-YAP interaction depends on upon the C-terminal PDZ-binding motif of YAP, which raises the question of whether Imp7 has a PDZ domain, or whether this is an atypical mode of interaction. It also raises the question of how *Drosophila* Yki could possibly interact with Imp7, since *Drosophila* Yki completely lacks the C-terminal PDZ-binding motif. Yet, the authors show that *Drosophila* Imp7 is essential for Yki-driven overgrowth. This doesn't quite square.
4. Importin-beta proteins normally bind to cargo via Importin-alpha, which acts as an adaptor protein to bind to the NLS sequences on the cargo. Where is the canonical NLS in YAP/TAZ and Yki? If there isn't one, isn't it likely that other YAP-binding proteins with an NLS might help mediate its binding to Importins?

Reviewer #2 (Remarks to the Author):

This is an exciting study about the nuclear import of the mechano-sensitive transcriptional regulator YAP. The authors first show that the nuclear transport receptor Imp-7 specifically changes its localization, nuclear or cytoplasmic, in response to different mechanical stimuli. No other NTR shows that behavior, so it is very specific. The authors further show that YAP is imported via Imp-7 and that the presence of YAP influences the binding of other Imp-7 cargo. This way the authors show two fundamentally new concepts in nucleo-cytoplasmic transport – a, that Imp-7s localization is mechanoresponsive, and, b, that YAP binding competes with other binders. ALL the experiments are carried out carefully and thoughtful, and I really cannot find a significant flaw. The Paper is also written elegantly and thoughtfully, and the storyline is easy to follow.

I enthusiastically recommend publication without further edits. This is very important work.

Reviewer #3 (Remarks to the Author):

Authors Garcia-Garcia et al propose in their manuscript “Mechanical control of nuclear import by Importin-7 is regulated by its dominant cargo YAP” a mechanism of nuclear transport involving mechanical cues. This work first uses mass spectrometry proteomics to identify candidate proteins that are differentially abundant in the nucleus vs cytosol under sparse vs confluent cell densities. Among the list of differential proteins is Importin-7 (Imp7), a nuclear transport receptor, and they validate that Imp7 increases abundance in the nucleus using microscopy. They then investigate the mechanism of Imp7 translocation during mechanical perturbation using many thorough imaging experiments.

I want to congratulate the authors for an intuitive, well-written research article. I found the rationale of their experiments clearly communicated and easy to follow, and I also appreciate that the interpretation of their results was concise and to the point. Overall, it made the manuscript a pleasure to read and I look forward to its publication in Nature Communications as a broadly interesting proposal of nuclear transport resulting from mechanical cues.

I have just a couple minor suggestions to improve the repeatability and clarity of the work, which are detailed below.

- Please include a supplemental table or description that defines which samples were labeled by which TMT channels, so that other researchers might reanalyze or mine the author’s mass spectrometry data.
- Table 1e is a bit difficult to parse. Could the authors move “Nuclear” and “Cytoplasmic” above the respective columns, so that “LD” and “HD” can be replaced with just “Low Confl” or “High Confl”? The other legends for this figure use “High Confluence” and “Low Confluence”, so reusing that language would make this subpanel more cohesive with the rest of the figure

Reviewers' comments are highlighted in blue.

Reviewer #1 (Remarks to the Author):

We want to thank the reviewer for her/his in-depth analysis and useful comments, which have allowed us to improve the manuscript. Specific points are addressed below.

“The manuscript investigates the role of the Importin-7 protein in nuclear import of YAP, and the ability of both proteins to be mechanically regulated in cell culture. The mechano-regulation of YAP has been the subject of intense investigation, and the current manuscript does not add significant new insight into the regulation of YAP.”

We agree with the reviewer that YAP has been studied extensively, being known that multiple proteins can regulate its activity by controlling its nuclear localization (Ma et al., *Annu. Rev. Biochem.* 88:577–604, 2019). However, the nuclear import factor directly responsible for its nuclear translocation has remained elusive. Here, we identify the main import factor responsible for YAP nuclear translocation, Imp7. We have screened for all known import factors and Imp7 is clearly the main one driving YAP nuclear entry (see below specific question related to this point). Imp7 is therefore a key new regulator of YAP, as it controls its nuclear accumulation. As a consequence, Imp7 regulates YAP function in gene expression (Fig. 5h-k) and organ growth *in vivo* (Fig. 6a, b).

In addition, we show that Imp7 is a highly mechanoresponsive protein, the first example of its kind in the pathways controlling nuclear transport. Furthermore, YAP is essential for Imp7 mechanoresponse and regulates the capacity of Imp7 to interact with other cargoes, resulting in the regulation of their nuclear accumulation. Therefore, our study not only focuses on YAP, but rather in the interplay between YAP and Imp7 and how this is important in the context of YAP function, mechanical cues, nuclear import regulation and signaling crosstalk. It is important to mention that with the new experiments suggested by this referee, we identify two essential regions in YAP needed to interact with Imp7 (being the NLS7 a new motif in YAP, see below and new Fig. S7), which provides important information for further studies attempting to block YAP nuclear entry and function.

“Rather, the novel results centre on Importin-7, which has long been known to participate in the nuclear import of multiple proteins, but has not been implicated in shuttling YAP through the nuclear pore. Thus, the identification of YAP as a cargo of Importin-7 is the main new finding.”

We agree that this is one of the main new findings of our work. This main discovery is complemented with two other important observations: first, Imp7 is mechanoresponsive, the first example of such behavior in the field of nuclear transport; and second, Imp7 is regulated

by its cargo YAP, which determines Imp7 localization in response to tension changes, and the ability to efficiently interact and import to the nucleus other cargoes (Smad3 and Erk2).

“siRNA silencing of Imp7/IPO7 inhibits nuclear accumulation of YAP/TAZ, although only partially (see Fig S6f!).”

The reviewer is right, the effect of Imp7 silencing on the nuclear accumulation of YAP/TAZ is partial, although highly reproducible and statistically significant (P value for the student's t-test is 2,14092E-11 for the mentioned figure). We performed several experiments to understand the reasons behind these results. Please see below the full response to this point in the answer to the major comment #1 (“1. Upon silencing of Imp7/IPO7, YAP is not excluded from the nucleus, being distributed at 1-1.....”).

Similarly, siRNA silencing of YAP/TAZ, or inhibition of MST1/2 kinases, modulates the subcellular localization of Imp7/IPO7, which suggests that the mechano-regulation of Imp7 may just be due to piggybacking on YAP/TAZ.

We agree with this view. The effect of YAP on Imp7 localization and function is not shared by other Imp7 cargoes (Fig. 8e, f), suggesting that although this effect may be due to piggybacking on YAP, it is specific for YAP, and most importantly, it has three functional consequences: i) regulates the localization of Imp7 (Fig. 8); ii) restricts the binding to other cargoes (Fig. 10 and S9m, n); and iii) regulates their nuclear accumulation (Fig. 9a,b and S9k, l). This mechanism allows mechanical cues to reach the nuclear import process mediated by Imp7 by providing a dominant cargo, YAP, resulting in signaling crosstalk.

Some further claims about competition between YAP/TAZ and other cargo of Imp7 are made, although the magnitude of the effect sizes are generally quite small in some of these experiments (e.g.: Fig S9l).

We agree that some of these effects are modest (about 24% increase on average in the cited experiment (Fig. S9l) or about 25% increase on average in the binding between Imp7 and Smad3 (Fig. 9b). In both cases the differences are highly reproducible and statistically robust (t test P value of 0,00019 and $1,14 \times 10^{-7}$ for the two siRNAs used in the cited figure S9l). These moderate effects reflect that the restrictions derived from YAP activation and binding to Imp7 are likely designed to fine tune, and not to shut down, other signaling pathways, as discussed in the text.

“The authors conclude that nuclear import ‘is not a mere passive shuttling system, but an additional regulatory layer in which a cargo (YAP) directs the nuclear transport receptor Imp7 and competes out other cargoes’. It is claimed that this is a ‘novel paradigm’.”

We have modified and tuned down this section following the specific recommendations indicated below.

“Overall, my feeling is that the authors are trying too hard to explain everything about mechanotransduction with their favourite molecule, Imp7. The manuscript would be greatly improved by (1) changing the title to simplify it (Importin-7 promotes YAP nuclear import);”

The proposed title (“Importin-7 promotes YAP nuclear import”) is quite clear and focused. However, we have to bear in mind that a significant portion of the manuscript is dedicated to the mechanoresponse of Imp7 (4 out of 10 main figures) and to the dominant effect of YAP on Imp7-driven import (2 out of 10 main figures). Thus, while the proposed title nicely narrows down the message, the current title covers the three main findings and concepts derived from the study, and we believe is more balanced, likely in agreement with the view of the other two referees, who enthusiastically recommended publication without changes or further *edits*.

“ (2) changing the abstract to make clear that the role of Imp7 is actually consistent with a passive shuttling system in which Imp7 may preferentially form a complex with YAP/TAZ over other cargoes; “

We agree that these changes will improve the message and we have made the corresponding modifications in the abstract. We have removed the sentence implying that the import process is an *active regulatory layer* and have reworded the final section to provide a general view regarding the consequences of our observations (marked in blue).

“(3) changing the results/discussion to moderate some of their strong interpretations.”

We agree that in some sections strong interpretations were made. We have made changes on those sections accordingly. See changes marked in blue in the text.

Major comments.

“1. Upon silencing of Imp7/IPO7, YAP is not excluded from the nucleus, being distributed at 1-1 nuclear/cytoplasmic ratio in the absence of Imp7, which raises the question of whether YAP/TAZ can also be a cargo for other Importins, just as Imp7 shuttles many different cargo.”

We performed several experiments to understand why YAP is being distributed at 1-1 nuclear/cytoplasmic ratio in the absence of Imp7. We hypothesized three possible explanations:

i) The cross-reactivity of the antibody (Santa Cruz 63.7) used to stain YAP, which recognizes also TAZ. To determine whether this could account for the observed partial effect, we used two alternative antibodies (Cell signaling technology 14074 and Santa Cruz H-9) that are more specific for YAP, as demonstrated (Dupont et al., *Nature*, 474, 179–183, 2011, for SC H-9) and confirmed in figure S5e. These new experiments rendered similar results to the ones obtained originally (direct comparison of all three antibodies is in new Fig. S5f). We also repeated the experiment shown before in Fig. S6f (now Fig. S5h-j) and cited earlier: (“*siRNA silencing of Imp7/IPO7 inhibits nuclear accumulation of YAP/TAZ, although only partially (see Fig S6f!).*”). The results were indistinguishable from those shown before with another antibody (compare

Fig. S5i and j). Thus, the antibody cross-reactivity with TAZ does not explain the partial effect observed upon Imp7 silencing.

ii) As suggested by the reviewer, a second possible explanation could be that other importins, in addition to Imp7, may contribute to the nuclear import of YAP. To test this possibility, we individually silenced all members of the nuclear transport receptor family involved in the import process (a total of 19). The knockdown efficiency was determined for all siRNAs (new Fig. S5m). As shown in figure 5f, Imp7 siRNA was the most effective treatment reducing the nucleo/cytoplasmic YAP ratio. Interestingly, a moderate effect of Importin- α 1, Importin- α 5 and Importin- α 8 was observed (new Fig. 5f). To understand whether these effects were due to the presence of parallel pathways taking YAP into the nucleus independently of Imp7, we tested whether the effects were accumulative when all positive hits were used simultaneously or in different combinations. Simultaneous knockdown of Importin- α 1, Importin- α 5 and Importin- α 8 did not induce an additive effect as the nucleo-cytoplasmic ratio was the same as the individual ones, suggesting that they were not parallel pathways (new Fig. 5g). When each of them (α 1, α 5 and α 8) were individually combined with Imp7, there was no additive effect either (new Fig. 5g). Similarly, simultaneous combination of all three importin α s and Imp7 rendered similar effects as Imp7 alone (new Fig. 5g). These results suggest that Importin- α 1, Importin- α 5 and Importin- α 8 are not parallel pathways to the Imp7-regulated pathway, rather this suggests that they act in a linear pathway. The consequences of knocking down α 1, α 5 and α 8 likely represent the effects on the import rate of other YAP regulators that control its phosphorylation (Zhao et al., *Genes Dev* 21, 2747-2761, 2007), stability (Sidor et al. *eLife* 2019;8:e48601), nuclear export (Ege, et al. *Cell Syst* 6, 692-708 e613, 2018), nuclear retention (Kim et al., *PNAS*, 117(24):13529-13540, 2020), cytoplasmic retention (Kanai et al., *EMBO J*, 24, 6778-6791, 2000) or other processes (Ma et al., *Annu Rev Biochem* 88, 577-604, 2019). An example of such process is described by Kim et al., (Kim et al., *PNAS*, 117(24):13529-13540, 2020). They describe MAML1, a nuclear YAP interacting protein, which contains a classical NLS recognized by Importin- α 1. When this protein is silenced, YAP is more cytoplasmic due to a reduction in the nuclear retention effect, produced by MAML1, on YAP. According to this rationale, knockdown of Importin- α 1, would reduce MAML1 in the nucleus, resulting in less nuclear YAP, yet this would not prove that Importin- α 1 drives YAP nuclear import. Therefore, to prove a direct effect of an importin on the nuclear import process, direct interaction between the cargo and the importin, and nuclear import assays with pure proteins are *sine qua non* experiments, as done for Imp7 (Fig. 4). In summary, the partial effect observed upon Imp7 silencing is likely not due to alternative import pathways directly importing YAP into the nucleus.

iii) A third possible explanation is that the silencing of Imp7 mRNA is not complete. Indeed, about 15-10% of the protein is still present upon knockdown (Fig. S5a) and the remaining protein may be able to import some YAP. Import receptors are in constant circulation, they take a cargo into the nucleus, release it, and return to the cytoplasm to import additional

cargoes. Therefore, the remaining Imp7 could account for some import, which may explain the partial effect. We attempted to delete the Imp7 gene (IPO7) by gene edition, however, we were unable to obtain knockout cell lines. Using the same technology and methodology we successfully obtained knockout cell lines for other genes (Echarri et al., *Nat. Commun.* 10:5828, 2019). Therefore, non-technical explanations likely explain this result. It is possible that Imp7 is an essential gene as it is needed for the nuclear import of important molecules, including histone H1, Erk, YAP, etc.

In summary, the partial effect on nuclear YAP when Imp7 is silenced is likely due to the remaining amount of Imp7 that is still present in the cell. We hope this answer is satisfactory as we have tested and excluded other possibilities as reasoned above.

“If so, it is difficult to see how the authors can conclude that Imp7 is a component of the Hippo pathway, or how Imp7 can be ‘the explanation’ for how YAP/TAZ is regulated by mechanical cues.”

Thanks for this appreciation. We agree that assigning Imp7 to the Hippo pathway was misleading and does not reflect the real function of Imp7, which is to induce the nuclear import of many different cargoes. For this reason, we have removed sentences related to this interpretation.

We also have toned down the sections or sentences where we provided an overstated view on the role of Imp7 in mechanically regulated YAP/TAZ.

“2. In *Drosophila*, silencing of Imp7 (*msk*) reduces imaginal disc size upon overexpression of UAS.Yki. This needs much further investigation.

We agree, this part was barely developed. We have now included several experiments that complement the results presented in the first version (i.e., Msk silencing prevents tissue overgrowth induced by expressed Yki). We addressed in full all the requested specific questions and provide additional experiments to understand the pathway in *Drosophila*. These results represent now two new figures (6 and S6) and a total of 20 panels. See below detailed responses to each point.

For example, what are the adult phenotypes of UAS.msk-RNAi in the eye and wing?

Flies expressing UAS.msk-RNAi under the control of apterus-Gal4 or engrailed-Gal4 were lethal, consistent with genetic studies using flies with deleted *msk* (Baker, et al., *Genetics*, 162, 285-296, 2002); this prevented the analysis of these organs in the adult flies. Previous studies have focused on the role of *msk* in the eye and wing biology. Vrailas et al., have shown that *msk* mutant cells survived in postmitotic territories in the eye (Vrailas et al., *Development* 133, 1485-1494, 2006). An independent study focused on the wing development and showed that Msk function is required for the growth of cells in the proliferating wing epithelium (Baker, et al., *Genetics*, 162, 285-296, 2002). These studies used genetic approaches to suppress Msk

expression in the developing phases of both organs. We performed a similar approach to express UAS.msk-RNAi in late stages of the development and restricted to a particular region/tissue. We expressed the UAS.msk-RNAi to knockdown *msk* under the *gmr*-Gal4 promotor, which restricts the expression to differentiated eye cells (Song et al., *Mol. Cell Biol.* 20, 2907-2914, 2000). As expected, *msk* silencing under these conditions did not produce any phenotype in the eye (see below Fig. R1 - "R" for reviewer inspection); consistent with a previous study, in which they showed that *msk* is not required for cell-type specification in the eye (Vrailas et al., *Development* 133, 1485-1494, 2006). While the expression of *msk-RNAi* under the control of engrailed-Gal4 (marked in red in Fig. R2) produced massive morphological alterations in the wing imaginal disc (Fig. R2), consistent with Baker et al., (Baker, et al., *Genetics*, 162, 285-296, 2002). Thus, partial reduction of Msk by RNAi is sufficient to induce cell death in epithelial wing imaginal disc during development. These results, shown in figures R1 and R2, are not included in the manuscript, as previous studies specifically oriented to understand Msk involvement in the wing and eye biology are already published (Baker, et al., *Genetics*, 162, 285-296, 2002; Vrailas et al., *Development* 133, 1485-1494, 2006). They could be included if required, although the scope of our study is not oriented to understand the eye and wing development mediated by Msk.

Figure R1.

Fig. R1 legend: Eye imaginal disc expression of *msk*-RNAi under the control of *gmr*-Gal4. The domain of expression is marked in red by *elav* staining. Active caspase 3 is shown in green. No evidence of apoptosis or morphological alterations are observed.

Figure R2.

Fig. R2 legend: Wing imaginal disc expression of *msk*-RNAi under the control of *engrailed*-Gal4 which is restricted to the posterior half territory of imaginal disc. The domain of expression is marked in red by the co-expression of UAS-Tomato. The cells included in *engrailed*-Gal4 normally occupy half of the total tissue and in this experiment, the posterior compartment is clearly reduced and show signs of apoptosis.

Is UAS.*msk*-RNAi able to prevent Yki nuclear localisation in *Drosophila*?

Yes, the new experiments (now shown in new figure 6c, d and S6c, d) show that UAS.msk-RNAi prevents Yki nuclear accumulation in *Drosophila*.

We used two different strategies rendering similar results. In epithelial wing imaginal discs overexpressing Yki (UAS-Yki), Yki staining with an antibody (kindly provided by Dr. Ken Irvine (Oh and Irvine, *Development* 135, 1081-1088, 2008)) showed that Yki was localized mostly in the cytoplasm in some cells while throughout the cytoplasm and the nucleus in other cells (consistent with Fletcher et al., *Development* 145, dev159467, (2018); Oh and Irvine, *Development* 135, 1081-1088, 2008; and Pan et al., *Development* 145, dev165712, 2018); in contrast, in Msk depleted cells Yki was mostly cytoplasmic in all cells (new Fig. 6c, quantified in d). Using a second approach, in transgenic flies expressing GFP-Yki, the GFP signal showed also a similar pattern to Yki, some cells showed mostly cytoplasmic signal, and cytoplasmic plus nuclear in other cells. In contrast, flies also expressing msk-RNAi showed GFP-Yki only in the cytoplasm (new Fig. S6c, quantified in d). Thus, Yki nuclear accumulation *in vivo* depends on Msk expression. This result is consistent with the effects of Msk silencing on Yki-induced tissue overgrowth (Fig. 6a, b).

And is the subcellular localisation of Imp7/Msk also regulated in the same way as Drosophila Yki? These are essential questions to answer.”

We now provide several new evidences *in vivo* and *ex vivo* suggesting that Msk subcellular localization is regulated in a similar manner to Yki; this is now shown in novel figures 6e, f and S6e-j. In flies, when endogenous Msk was cytoplasmic, Yki was also cytoplasmic, while in cells where Msk was cytoplasmic and nuclear, Yki showed the same distribution (new Fig. 6e), resulting in a strong correlation of both staining (Pearson’s coefficient of 0.85 ± 0.04 SD). Similar results were obtained in transgenic flies expressing YFP-Msk (Liu and Geisbrecht, *Dev. Biol* 359, 176–189, 2011), where the YFP-Msk signal and the staining with anti-Msk showed a similar pattern to endogenous Yki (new Fig. 6f, Pearson’s coefficient of 0.912 ± 0.035 SD, for Yki and Msk staining). To gain more insight into the mechanisms that regulate Msk subcellular localization and whether they were similar to the mechanisms governing Yki subcellular localization, we used *Drosophila* S2 cells. These cells grow in suspension and semi-attached conditions; under these conditions Msk was mostly cytoplasmic, like Yki (new Fig. S6e, f). However, induction of cell spreading, which induces activation of tension-controlling pathways (Tee, et al., *Biophys J* 100, L25-27, 2011), induced Msk nuclear accumulation, similar to Yki (new Fig. S6e, f). Interestingly, actin polymerization disruption with cytochalasin D prevented the nuclear accumulation of Msk (novel Fig. S6g, h), similar to the effect observed in Yki function in S2 cells (Sansoles-García et al., *EMBO J.* 30, 2325–2335, 2011). Furthermore, fully spread S2 cells growing at low density showed nuclear Msk, while cells growing at high density displayed cytosolic Msk (new Fig. S6i, j), as occurs to Yki (Pan et al., *Development* 145,

dev165712, 2018) and Imp7 localization (Fig. 1c, d). Thus, in terms of subcellular localization, Msk behaves similar to Yki, Imp7 and YAP/TAZ (Fig. 1 and 3 and Dupont et al., *Nature* 474, 179-183, 2011).

To further characterize the interplay between Msk and Yki we studied the interaction between these proteins. This is now shown in new Fig. 6g, h and S6k, l. Briefly, Msk and Yki interact in cells in a cell spreading and Yki C-terminus dependent manner. The involvement of the C-terminus of Yki is explained below in detail (see point 3).

“3. The Imp7-YAP interaction depends on upon the C-terminal PDZ-binding motif of YAP, which raises the question of whether Imp7 has a PDZ domain, or whether this is an atypical mode of interaction.”

Based on the current domain search engines, such as SMART (Schultz et al., *Nucleic Acids Res.* 28(1):231-4, 2000), it has been shown that there are about 500 proteins containing a PDZ domain in the human genome (Liu and Fuentes, *Int Rev Cell Mol Biol*, 343: 129–218, 2019). However, based on this methodology, Imp7 does not contain a PDZ domain. Whether Imp7 has a PDZ domain in its 3D structure, based on the crystal (not available) or based on predictions (such as that developed by *AlphaFold*, Jumper et al., *Nature* volume 596, 583–589, 2021), is currently unknown. Therefore, with the current information, the interaction of Imp7 with YAP may be an atypical mode of interaction, especially considering that there are other regions within YAP that are important for the interaction to take place (please see below point 4).

“It also raises the question of how *Drosophila* Yki could possibly interact with Imp7, since *Drosophila* Yki completely lacks the C-terminal PDZ-binding motif. Yet, the authors show that *Drosophila* Imp7 is essential for Yki-driven overgrowth. This doesn't quite square.”

To understand this apparent discrepancy, we first empirically tested whether Yki could interact with Msk and whether the interaction was dependent on the C-terminus last amino acids of Yki, like in the case of the YAP-Imp7 interaction. As shown in new figure 6g, h, Yki interacts with Msk, and interestingly, the interaction is fully dependent on its C-terminal last 7 amino acids. Thus, the C-terminal region of YAP and Yki are necessary to interact with Imp7 and Msk, respectively. This raises the possibility that the C-terminus of YAP and Yki may share some similarities. It is true that a standard alignment of Yki and human YAP (using CLUSTAL O (1.2.4)) does not align the C-terminus of both proteins, therefore this leads to the conclusion that in Yki there is not an equivalent region to the PDZ-binding motif of YAP. Interestingly, the original study that identified Yki (Huang et al., *Cell* 122, 421–434, 2005) performed an alternative alignment between Yki and YAP, in which the C-terminus of YAP (including the PDZ-binding motif) co-aligned with the C-terminus of Yki (see below Fig. R3a) (Huang et al., *Cell* 122, 421–434, 2005). Supporting this alignment, a recent study focusing on the evolution of the YAP

protein also co-aligned Yki and YAP in the PDZ-binding motif region (Ikmi et al., *Mol Biol Evol*, (6):1375-90 2014, -see below Fig. R3b-). This study suggested that the PDZ-binding motif is conserved in many organisms, including the cnidarian *Nematostella vectensis*, which is evolutionary distant from human YAP and *Drosophila* Yki (Ikmi et al., *Mol Biol Evol*, (6):1375-90 2014). Another study, however, did not interpret, based on a similar alignment, that the C-terminus of Yki was similar enough to be considered a *bona fide* PDZ-binding motif (Oka and Sudol, *Genes Cells*, 14(5):607-15, 2009), as only two out of five amino acids of YAP are conserved. Whether the last 7 amino acids of Yki can be categorized as a PDZ-binding motif is currently unclear, but at least with respect to binding to Msk, we can state that the last 7 amino acids of Yki are necessary for the interaction to occur (new Fig. 6g, h), as in the case of YAP with its last 5 amino acids (new Fig. S7a, d).

Figure R3

a)

CLUSTAL O(1.2.4) multiple sequence alignment

```

hYAP_c-term  ----- 0
Yki_FL      MCACLIAKIILCSFRLYTISAFYMLTTMSASSNTNSLIEKEIDDEDMLSPIKSNLVRV 60

hYAP_c-term  -----QPPPLAPQSP---QGGVMGGS 18
Yki_FL      NQDTDDNLQALFDSVLNPGDAKRPLQLPLMRKLPNSFFTPPAPSHSRANSADSTYDAGS 120
                          ** :* :. .**

hYAP_c-term  NSNQ-----QQMRLQQLQM-----EKERLRKQQL-----LR 47
Yki_FL      QSSINIGNKASIVQPPDGGQSPIAAIPQLQIQSPQHSLAIHHSRRARSSPASLQQNYNVR 180
      :* .      : ***:        .:. * . .* :.*

hYAP_c-term  Q-----AMRNINPSTANSKPCQELALRSQLPTLEQDGGTQNPVSSPGMS-QELRTMTT 99
Yki_FL      ARSDAAAANPNANPSSQQPA-----GPTFPENSAQEFPSGAPASSAIDLAMNT 231
                          * ***: :.*           **: :... : * :.* :.* :.*

hYAP_c-term  NSS-DPFLNSGTYHSRDESTD-----SGLSMSSYSVPRTPDF--LN----- 138
Yki_FL      CMSQDIPMSMQTVHKKQRSYDVISPIQLNRQLGALPPGWEQAKTNDGQIYYLNHHTTKSTQ 291
      ** :. * * :... * *           *       ... : * *       **

hYAP_c-term  -----SVDEMDTGDTINQSTLPSQQNRFPDYLEAIPGTNVDLGT 177
Yki_FL      WEDPRIQYRQQQILMAERIKQNDVLQTTKQTTTSTIANNLGPLPDGWEQAVTESGDLYF 351
                          . * ::* . . ** : . . : ** * . **

hYAP_c-term  L-----EGDGMNI--EGEELMP--SLQEALSSDILNDMESVLAATKLDKESF 220
Yki_FL      INHIDRTTSWNDPRMQSGLSVLDCPDNLVSSLQIEDNLCSNLFNDAQAIVNPPSSHKPDD 411
      :                 .*:.: :*: ::: *:*::** ::: . .* .

hYAP_c-term  LTWL--- 224
Yki_FL      LEWYKIN 418
* *

```

b)

[REDACTED]

Fig. R3 legend: a) Alignment of YAP and Yki. Sequences were aligned using again CLUSTAL O(1.2.4). To produce the alignment, we used human YAP (NP_001123617, starting in amino acid 281) and Yki described by Huang et al. (H0RNA4 (H0RNA4_DROME) from Uniprot, 418 amino acids). **b)** A snapshot of the YAP/TAZ family alignment shown in Ikmi et al., *Mol Biol Evol*, (6):1375-90, 2014, is shown.

“4. Importin-beta proteins normally bind to cargo via Importin-alpha, which acts as an adaptor protein to bind to the NLS sequences on the cargo. Where is the canonical NLS in YAP/TAZ and Yki? If there isn't one, isn't it likely that other YAP-binding proteins with an NLS might help mediate its binding to Importins?”

YAP/TAZ and Yki do not contain a canonical NLS, at least based on their primary sequence analysis using *Uniprot*, *Prosites* or *cNLS Mapper* search engines, in agreement with other analyses (Kim et al., *PNAS*, 117(24):13529-13540, 2020; and Kofler et al., *Nat. Comm.* 9:4966, 2018). However, for a protein to be imported into the nucleus, there are multiple types of non-canonical NLSs that act independently of canonical NLS and the Importin- α/β 1 system (Soniati et al., *Biochem. J.* (2015) 468, 353–362). Imp7, which belongs to the importin β subfamily, can drive the nuclear import of cargoes autonomously (as in the case of YAP, Fig. 4d, and Jakel, et al., *EMBO J.*, 17, 4491-4502, 1998; Fassati et al., *EMBO J.*, 14, 3675-3685, 2003) or with the aid of Importin- β 1 (Jakel, et al., *EMBO J.* 18, 2411-2423, 1999) and does not require importin α s nor a canonical NLS to drive nuclear translocation of cargoes (Jakel, et al., *EMBO j.*, 17, 4491-4502, 1998). In the case of the *Drosophila* Msk, it is known that it cannot recognize a canonical NLS (Xu et al., *J Cell Biol*, 178, 981-994, 2007). In the case of the YAP cargo, Imp7 is sufficient to drive its nuclear import, without the aid of any other importin (Fig. 4d-f). Some importin β members recognize specific sequences that act as non-canonical NLSs. However, in the case of Imp7, it is not yet clear which NLS recognizes (Soniati et al., *Biochem. J.* 468, 353–362, 2015).

Two studies have proposed potential NLSs for Imp7. In a recent study by Panagiotopoulos et al., (Panagiotopoulos et al, *BBA-General Subjects* 1865 (2021) 129851), the authors proposed the following NLS consensus sequence: EKRKI(E/R)(K/L/R/S/T), named NLS7. A different study proposed that the S/T-P-S/T consensus sequence acts as an NLS for Imp7 (Chuderland et al., *Mol. Cell* 31, 850–861, 2008). YAP contains sequences resembling those proposed NLSs. In addition, YAP has two regions with basic amino acids (RK-RK) and we also selected those regions as potential NLSs, as basic amino acids are frequently part of NLSs (Soniati et al., *Biochem. J.* 468, 353–362, 2015). Thus, we independently mutated all these three different potential NLSs and determine their effect on binding to Imp7 and on their nuclear localization. The results (new Fig. S7c-g) suggest that the putative NLS7 in YAP is required to interact with Imp7 and to localize YAP to the nucleus. The RK-RK region (named

RK²) moderately affects binding to Imp7 and nuclear accumulation, while the TPT region moderately affects binding to Imp7 but has no clear effect in nuclear accumulation (new Fig. S7c-g). In summary, we have identified in YAP two essential regions to interact with Imp7 and to localize it to the nucleus: the NLS7 (amino acids 314-320) and the *PDZ-binding motif* (amino acids 500-504, in this case it was previously known to affect YAP nuclear localization, but it was unclear why; Oka, T. & Sudol, M. *Genes Cells* 14, 607-615 (2009)). In addition, we identified serine 127 as an amino acid essential to modulate the binding to Imp7: when phosphorylated, YAP does not bind Imp7 (new Fig. 7i, j and S7b, d), which prevents its nuclear accumulation as described before (see Zhao et al., *Genes Dev* 21, 2747-2761, 2007). New figures S7d & e summarize all the YAP mutants used in this study and their binding to Imp7.

A study in *Drosophila* proposed that the region 1-55 of Yki “may serve as an NLS” (Wang et al., *J. Biol. Chem.* 291, 7926–7937, 2016); however, the evidences supporting this claim were scarce and import assays to demonstrate the direct involvement of Importin- α 1 were not conducted (see below for a longer rationale on the importance of these assays). Furthermore, an independent group attempted to prove that this region in YAP also contained an NLS, but after conducting sequence analysis and mutagenesis experiments, they concluded that there was not an NLS in that region (Kim et al., *PNAS*, 117(24):13529-13540, 2020). Within the region 1-55 of Yki, Wang et al., showed that Arginine 15 was essential for nuclear accumulation but this residue is not present in YAP and YAP lacking the first 58 amino acids is fully competent in activating YAP target genes (Kim et al., *PNAS*, 117(24):13529-13540, 2020). For these reasons we did not pursue searching putative NLSs in this region.

The possibility raised by the reviewer (“isn't it likely that other YAP-binding proteins with an NLS might help mediate its binding to Importins?”) is possible, but this has not been proven yet, to the best of our knowledge. There are two interesting studies showing that two independent proteins containing canonical NLS regulate YAP/TAZ nuclear localization (MAML1/2 proteins, Kim et al., *PNAS*, 117(24):13529-13540, 2020; ANKHD1/ANKRD17 proteins, Sidor et al. *eLife* ;8:e48601, 2019). However, these studies have not demonstrated that these proteins form a complex with YAP/TAZ and Importin- α , nor that Importin- α mediates the nuclear import of YAP/TAZ, bridged by those proteins, using nuclear import assays. These studies showed that those proteins (MAML1/2 or ANKHD1/ANKRD17) regulate the amount of YAP in the nucleus and this can occur by nuclear retention, as shown in the case of MAML1/2 (Kim et al., *PNAS*, 117(24):13529-13540, 2020) or by other mechanisms (Sidor et al. *eLife* ;8:e48601, 2019).

It is important to differentiate between: 1) To regulate the amount of nuclear YAP, and 2) To drive the nuclear import of YAP. The amount of nuclear YAP is regulated in many ways: by affecting its phosphorylation (Zhao et al., *Genes Dev* 21, 2747-2761, 2007), stability (Sidor et al. *eLife* 2019;8:e48601), nuclear export (Ege, et al. *Cell Syst* 6, 692-708 e613, 2018), nuclear retention (Kim et al., *PNAS*, 117(24):13529-13540, 2020), cytoplasmic retention (Kanai et al., *EMBO J*, 24, 6778-6791, 2000), and by many proteins (Ma et al., *Annu Rev Biochem* 88, 577-604, 2019). When these regulatory circuits are manipulated, an effect in the amount of nuclear

YAP can be observed. However, this doesn't mean that the effect is due to an alteration in the nuclear import process *per se*. Even when an importin is silenced and this impacts on the amount of a protein in the nucleus, it doesn't mean that it is directly involved in the import process, as its effect on other proteins regulating the protein of interest may be responsible, i.e., an indirect effect. To unambiguously state that an importin drives the nuclear translocation of a cargo, in this case YAP, a necessary experiment must be performed: a nuclear import assay (either by nuclear import assay on permeabilized cells or a FG-particle assay, both of which we show in figures 4d-f for Imp7/YAP). As stated earlier, in the mentioned studies (Sidor et al., and Kim et al.), the authors did not provide any direct evidence that YAP was being imported into the nucleus by Importin- α 1 or any other protein. In summary, there are no evidences that YAP/TAZ or Yki have a canonical NLS, and although it is theoretically possible that other proteins containing a canonical NLS bridge YAP/TAZ or Yki to Importin- α/β 1, there are no reports showing this. Our results obtained in the screening of importins, especially the data silencing multiple importins, suggest that there isn't an alternative nuclear entry route for YAP directly mediated by other importins, and parallel to Imp7 (new Fig. 5f, g); please see our response to point 1 for additional rationale. These ideas are discussed in the manuscript (page 20).

Reviewer #2 (Remarks to the Author):

"This is an exciting study about the nuclear import of the mechano-sensitive transcriptional regulator YAP. The authors first show that the nuclear transport receptor Imp-7 specifically changes its localization, nuclear or cytoplasmic, in response to different mechanical stimuli. No other NTR shows that behavior, so it is very specific. The authors further show that YAP is imported via Imp-7 and that the presence of YAP influences the binding of other Imp-7 cargo. This way the authors show two fundamentally new concepts in nucleo-cytoplasmic transport – a, that Imp-7s localization is mechanoresponsive, and, b, that YAP binding competes with other binders.

ALL the experiments are carried out carefully and thoughtful, and I really cannot find a significant flaw. The Paper is also written elegantly and thoughtfully, and the storyline is easy to follow.

I enthusiastically recommend publication without further edits. This is very important work."

We sincerely thank this reviewer for her/his enthusiastic and positive comments. This is a strong motivation for us to continue this line of research.

Reviewer #3 (Remarks to the Author):

"Authors Garcia-Garcia et al propose in their manuscript "Mechanical control of nuclear import by Importin-7 is regulated by its dominant cargo YAP" a mechanism of nuclear transport

involving mechanical cues. This work first uses mass spectrometry proteomics to identify candidate proteins that are differentially abundant in the nucleus vs cytosol under sparse vs confluent cell densities. Among the list of differential proteins is Importin-7 (Imp7), a nuclear transport receptor, and they validate that Imp7 increases abundance in the nucleus using microscopy. They then investigate the mechanism of Imp7 translocation during mechanical perturbation using many thorough imaging experiments.

I want to congratulate the authors for an intuitive, well-written research article. I found the rationale of their experiments clearly communicated and easy to follow, and I also appreciate that the interpretation of their results was concise and to the point. Overall, it made the manuscript a pleasure to read and I look forward to its publication in Nature Communications as a broadly interesting proposal of nuclear transport resulting from mechanical cues”.

We sincerely thank the reviewer for such positive and encouraging comments.

“I have just a couple minor suggestions to improve the repeatability and clarity of the work, which are detailed below.”

- Please include a supplemental table or description that defines which samples were labeled by which TMT channels, so that other researchers might reanalyze or mine the author’s mass spectrometry data.”

Thanks for pointing this out, we apologize for not including this information. This is now included in a table in figure S1k.

“- Table 1e is a bit difficult to parse. Could the authors move “Nuclear” and “Cytoplasmic” above the respective columns, so that “LD” and “HD” can be replaced with just “Low Confl” or “High Confl”? The other legends for this figure use “High Confluence” and “Low Confluence”, so reusing that language would make this subpanel more cohesive with the rest of the figure”.

Thanks for pointing this out, we apologize for not being consistent in the nomenclature. This is now modified accordingly.

REVIEWERS' COMMENTS

Reviewer #1 (Remarks to the Author):

The manuscript is significantly improved, and could now be published in NCOMMS.

Point-by-point rebuttal letter

NCOMMS-21-15055A

Reviewer's comments are in *blue*

Reviewer #1 (Remarks to the Author):

"The manuscript is significantly improved, and could now be published in NCOMMS."

We want to thank the reviewer for her/his comments. Her/his analysis allowed us to significantly improve the manuscript.